# Tribocatalysis Induced Carbon-Based Tribofilms—An Emerging Tribological Approach for Sustainable Lubrications

**Khai K. Huynh** [1], **Sang T. Pham** [1,2,*], **Kiet A. Tieu** [1,*] **and Shanhong Wan** [3]

1 School of Mechanical, Materials, Mechatronic and Biomedical Engineering, University of Wollongong, Wollongong, NSW 2522, Australia; kkh981@uowmail.edu.au

2 School of Chemical and Process Engineering, School of Chemistry, and Bragg Centre for Materials Research, University of Leeds, Woodhouse Lane, Leeds LS2 9JT, UK

3 State Key Laboratory of Solid Lubrication, Lanzhou Institute of Chemical Physics, Chinese Academy of Sciences, Lanzhou 730000, China; shwan@licp.cas.cn

* Correspondence: t.s.pham@leeds.ac.uk (S.T.P.); ktieu@uow.edu.au (K.A.T.)

**Abstract:** To comply with the high demand for efficient and sustainable lubrications, carbon-based tribofilms and/or nanomaterials have emerged as a potential solution that can resolve the current major shortcomings of phosphorus- and sulphur-rich tribofilms and protective coatings. Although their employment is still in the early stages of realization and research, these tribofilms receive significant interest due to their capability to continuously and in situ repair/replenish themselves during sliding, which has been an ultimate goal of all moving mechanical systems. Structurally, these tribofilms are complex and predominantly amorphous or disordered with/without graphitic domains (e.g., graphene/graphite, onion-like carbon, etc.). Chemically, the compositions of these tribofilms vary significantly with environments, conditions, and material precursors. Yet, the structural properties of carbon-based tribofilms remain largely ambiguous, which precludes a full understanding of the mechanisms underlying the formation and lubrication performance. This review will summarize the current state-of-art research about the in situ carbon-based tribofilms that have been published since the pioneering works. Particularly, this work will highlight the recent approaches to generate these tribofilms, their associated lubrication performance, current understanding of the formation mechanics, common analytical approaches for these tribofilms, and the compatibility of these tribofilms with other additives. Together, the overall outlooks will be drawn, demonstrating the knowledge gaps and proposing further investigation tactics to tackle these emerging issues.

**Keywords:** carbon-based tribofilms; sustainable lubrications; tribocatalysis; friction; wear

## 1. Introduction

The growth of transportation has caused significant concerns about environmental impact and energy consumption. The total loss of energy in internal combustion engines (ICE) results in 28% of the annual $CO_2$ emissions around the world [1,2]. Recently, electric vehicles (EV) and hybrid vehicles (HEV) have emerged as sustainable technologies to optimize energy efficiency, which are predicted to halve the energy losses due to friction [3]. EV and HEV technologies are still in the progress of development, and the design of electric motors, batteries, transmission materials and fluids largely depends on the manufacturer. Despite the ongoing progress of EVs and HEVs, the current road transport system, especially with heavy-duty vehicles, is still based on ICE [4]. The recent development and employment of heavy-duty vehicles with EV and HEV technologies have been proposed [2,5,6], but these alternatives start from a very low base and still face significant barriers to complete their unlimited expansion [7]. A total of 85–90% of road transport energy is still expected to come from ICE-powered vehicles even by 2040 [7]. The impacts of tribological contacts in ICE on the economy and environment are still an unabated issue for the next few decades.

Improving the energy efficiency in conventional ICE vehicles is critical to reduce $CO_2$ emissions and improve the economy. The energy losses and surface degradations in ICE predominantly originate from the micro-asperity contacts between sliding surfaces. There are two common strategies to overcome the adverse effects of these solid contacts, including (i) optimizing the lubricant packages; and (ii) improving the contacting interface, by surface texturing or by the deposition of functional lubricous coatings on selected component surfaces [8–10]. Both approaches have their unique advantages and are often studied complementarily to achieve synergistic lubrication effects. Sustainable lubrication should reduce fuel consumption, curb carbon emissions, and deliver positive economic impacts.

Carbon-based material has been a long-standing research area in both approaches of lubricating solid contacts. This is a sustainable class of materials that show outstanding performance in controlling friction and wear in modern vehicles. On the lubricant side, carbon-containing additives, such as carbon nanotubes, graphene, or amorphous carbon, can be incorporated into lubricant packages to enhance their performance [11–16]. The lubrication mechanisms of these additives mostly come from their high strength, high hardness, easy shear property of layered structures, and/or incommensurate contact among the inner graphitic units [17–21]. Meanwhile, carbon-based coatings, i.e., diamond-like carbon (DLC) coatings, are commonly used to modify the contacting interface and improve performance due to their hard and self-lubricating properties [22,23]. While carbon-based solid lubricant additives often require complicated chemical synthesis and surface modifications to incorporate into the lubricant packages [24], carbon-based coatings face a major issue of limited thickness that results in eventual wear-out after a limited time of operation [25]. Carbon-based materials have far-reaching implications in the future lubrication but extensive research should be conducted to bring these materials to reach the final practical applications.

Recently, the in situ creation of carbon-based materials, either as nanoparticles or nanotribofilms, at the sliding interface is a subject of ongoing research and development in tribological perspectives [26]. Based on this approach, nanocarbon materials can be self-generated, meaning they can continuously form or repair themselves in response to wear or damage, providing continuous protection to the metal surfaces. This approach is a potential alternative for replacing the use of conventional extreme pressure (EP) antiwear additives (e.g., zinc dialkyldithiophosphate—ZDDP) to adapt to new emissions regulations [27–29] and a current trend of using a low-viscosity oil lubricant that weighs a burden on the wear protection in engine and transmission parts [30,31]. It is expected that tribochemical and/or mechanochemical reactions often take place between the lubricants and sliding surfaces under subsequent tribological contact, where high frictional heating, contact pressure, and stress-shearing occur. Carbon-based tribofilms can be formed from a variety of carbon precursors under these events [26]. During the sliding process, the presence of these carbon-rich layers on the contacting surfaces will minimize the contact stresses between sliding asperity peaks [32]. This not only limits frictional issues but also protects the solid surfaces against severe wear [33]. There are two current approaches to create nanocarbon materials in situ on sliding surfaces, including the utilization of carbon-precursor lubricants/additives [34,35] and the employment of catalytic materials to promote tribocatalysis [26]. It has been demonstrated that the structures and properties of nanocarbon materials vary widely depending on the conditions they are created [26]. The pioneering research of forming carbon-based materials in situ was conducted by Erdemir et al. [36], where amorphous carbon and onion-like carbon were formed from the polyalphaolefin oil by nanocrystalline catalytic coatings. In the following year, the use of oil-soluble carbon-precursor additives, e.g., cyclopropanecarboxylic acid (CPCa), was conducted to form polymeric amorphous carbon-based tribofilms [34,37–40]. Currently, the use of catalytically active nanoparticles was also proposed to control the formation of amorphous carbon-based tribofilms [41,42]. A number of studies regarding nanocarbon tribofilms have significantly arisen since 2016 (Figure 1). However, rich literature has largely relied on systematic studies on the tribological performance, while the structural

and chemical properties of these tribofilms has not yet been well understood. It can be seen that almost 70% of published works indicate the disordered/amorphous structures of the nanocarbon tribofilms without any details on any turbostratic or parasitic characters in the tribofilms.

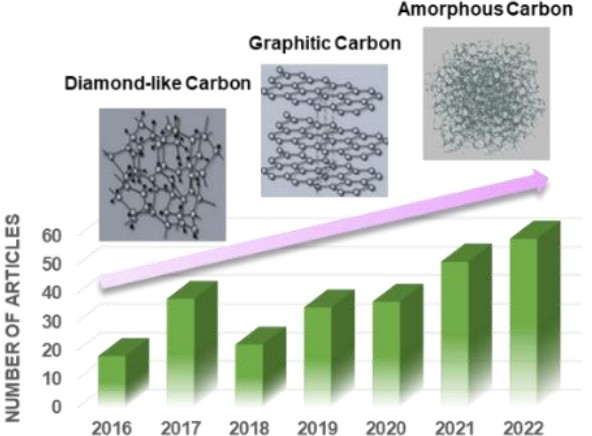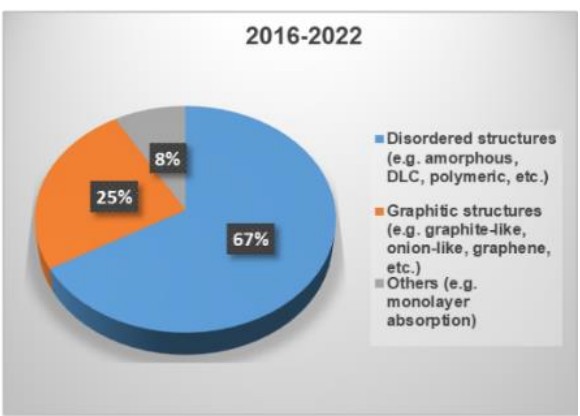

**Figure 1.** Progress on the research of carbon-based tribofilms formed at the sliding interface. The data has been collected and summarized from Scopus Database—Elsevier from 2016 to 2022.

The majority of conclusions about the lubrication and formation mechanisms as well as chemical structures of carbon-based tribofilms are often drawn from macroscopic observation of the wear surfaces, low-spatial-resolution analysis, and crystalline-dependent diffraction analysis. Such information is still controversial in tribological research. Answering these bottlenecks can open a new pathway to optimize the stability and compatibility of carbon tribofilms with other lubricant components, as well as controlling long-term performance and durability under different operating conditions. In the following section, we will review the state-of-the-art studies in controlling the formation of the carbon-based tribofilms at the sliding interface in relation to the lubrication performance, as well as discussing the up-to-date fundamental mechanisms that govern the formation of these films. We will also review the current controversy in characterizing carbon-based tribofilms and the compatibility of carbon tribofilms with other additives, before proposing our perspectives in elevating this emerging field of research.

## 2. Carbon-Based Thin Films and Their Lubrication Performance

Carbon-based films are widely used in tribological systems to reduce friction and wear. These films can be polymeric, graphitic, or diamond-like, which are commonly prepared by chemical or physical methods. Publications about carbon-based films date back to the 1950s, when a diamond-like carbon (DLC) coating comprising an amorphous network was first introduced by Schmellenmeier [43]. Since then, numerous kinds of DLC coatings with numerous characteristics have been applied in various technical fields, such as hard disks, space technology, biological applications, chemical pumps, and food processing [44]. The structural characteristic of the DLC coatings can be flexibly tuned to produce graphite-like coatings [45]. Apart from DLC and graphite-like coatings, some studies in the early 1960s also described the appearance of high-molecular-weight products, or carbon-based friction polymers, formed in sliding metal systems lubricated by hydrocarbons [46]. In very recent studies, some fundamental works have sought to explore the mechanisms of friction polymers based on the context of mechanochemistry [47]. In general, carbon-based films can take several forms, and their tribological performance in terms of friction and wear are discussed and reviewed in the following parts.

### 2.1. Diamond-Like Carbon Films

In automotive applications, the use of DLC has been popular since the 1990s. It began with Volkswagen's diesel injection system, and later spread wide to various components like valve shafts, finger followers, piston skirts, tappets, piston rings, and piston pins [48,49]. Their properties are significantly attributed to their portions of carbon atoms' $sp^{2-}$ (C=C) and $sp^{3-}$ bonds (C–C and C–H). The DLC coatings with a high $sp^{3-}$ bonding portion (highly diamond-like) structure will be harder and exhibit better wear resistance in the tribological system [50]. Meanwhile, high $sp^{2-}$ bond portion DLC films as graphitic films will be softer, and result in better antifriction performance compared to the high $sp^{3-}$ bond portion [51]. Moreover, the concentration of hydrogen content in these carbon-rich coatings can also exert a certain effect on their tribological performances [52]. Two important groups of DLC film structures comprise the nonhydrogenated networks with the amorphous (a-C) and tetrahedral amorphous carbon (ta-C), and the hydrogenated one with the significantly higher hydrogen content like a-C:H. Their $sp^{2-}$, $sp^{3-}$, and hydrogen (H) contents can be described by the ternary phase diagram in Figure 2a. The "no film" area on the right-hand side stands for the formation of polyacetylene $(CH)_n$ and/or polyethylene $(CH_2)_n$, where only hydrocarbon molecules (instead of the carbon-based networks) are promoted [53].

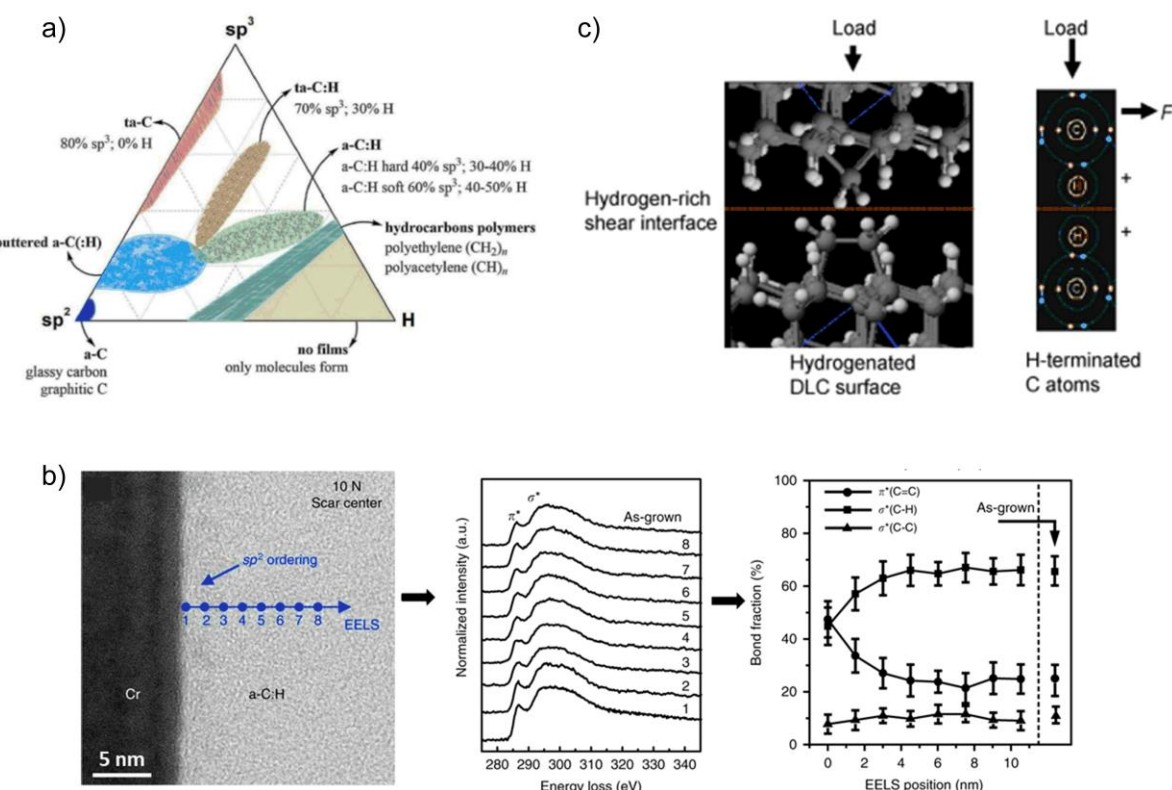

**Figure 2.** (**a**) Ternary phase diagram of common DLC tribofilms obtained from ref [54]. (**b**) Typical example of "friction-induced graphitization" mechanism [55] and (**c**) passivation mechanism [56].

There are two mechanisms that are widely postulated as the main antifriction property of DLC tribofilm, including the "friction-induced graphitization" mechanism and passivation mechanism. Particularly, graphitization is the transformation process of the carbon-based films from amorphous to graphitic ordered form, with a higher $sp^2$ portion in bonding configurations [57]. Then, the "friction-induced graphitization" mechanism is defined as the process where the rise of the $sp^2$-bond portion at the sliding interface due to sliding friction [55] (Figure 2b) would result in a graphitized tribofilm with stable and low induced friction corresponding to the lubricious nature of graphite. Meanwhile, the passivation mechanism is the procedure where the majority of carbon with free dangling σ-

bonds are passivated or saturated by hydrogen terminations as passivating species to form a nonreactive surface (Figure 2c) [17]. The estimated content of hydrogen in DLC coating that can protect the interface areas from adhesive interactions due to the reformations of dangling carbon atoms is around 40 at% [52,58]. From the atomic-scale point of view, this mechanism is attributed to the charge density redistribution and the positive hydrogen protons' repulsive forces, which suppress the adhesion interactions and produce low-friction outcomes [56]. It can be seen that the hydrogen atoms play a significant role in the DLC passivation mechanism. Its presence in the matrix of amorphous carbon would soften the structure by increasing the void density and/or releasing the intrinsic pressure [17].

Among the aforementioned film structures, the a-C:H coating has been highlighted as the only DLC film that can achieve a lower than 0.01 friction outcome [56,59,60]. This can be explained by its unique structure with sufficient $sp^2$ and H contents, which can effectively process both "friction-induced graphitization" and passivation mechanisms during the sliding process [55]. In contrast, for a high $sp^3$ portion and hydrogen-deficient coating in ta-C or nanocrystalline diamond, the long continuous graphitization process without a suitable saturation can easily result in a high-friction outcome and catastrophic adhesive wear. Furthermore, another major issue with the $sp^3$-rich DLC films that has significantly restricted their wide employment is their low absorption strength to the base substrate [61]. This can be explained by the association of the $sp^3$ bonding with the compressive stresses of the DLC film [61–64]. To reduce the residual stress and improve the films' desired mechanical properties, the introduction of functional metallic atoms as doping elements and/or interlayers is recognized as one of the most effective approaches, as demonstrated by numerous studies [22,23,65]. It has been widely acknowledged that the graphitic degree of carbon-based tribofilms can be promoted by providing the carbon matrix with such catalytic metals as Cu, Re, Pt, Pd, and Ni during the manufacturing process [57,62]. Meanwhile, to increase the DLC coating's $sp^3$ bonding portion without weakening its adhesion, the use of Cr or Al interlayers could be considered [62,66].

According to Zhou et al. [65], many techniques to deposit the DLC coating on a target substrate have been introduced, with plasma-enhanced chemical vapour deposition (PECVD), physical vapour deposition (PVD), and chemical vapour deposition (CVD) categorized as the most common ones. Among these, magnetron sputtering as a PVD approach has been widely employed in many industrial fields. This approach utilizes energetic ions produced from the discharge plasma to bombard the target surface [22]. The popularity of the magnetron sputtering method can be explained by the diversity of power supplies like direct current, radiofrequency, etc., which can be monitored to tailor the desired carbon-based tribofilm structures [65]. Moreover, it is clear from Al Mahmud et al. [22] that such deposition techniques as plasma-activated chemical vapour deposition, plasma immersion ion implantation and deposition, or ion beam deposition have also received much interest due to their flexibility. In general, their working temperatures usually range between subzero and 400 °C. Furthermore, etching time, bias voltage, and carbon-precursor gases (like acetylene [65], methane [67], butane [68], etc.) are also important working parameters that can be controlled in order to achieve the target carbon-based coatings. Detailed advantages and disadvantages of each deposition approach can be found in the work performed by Hainsworth et al. [69].

### 2.2. Carbon-Based Tribofilms Produced by Catalytic Coatings

Although DLC coatings have already proven their lubricating functions in practice [48,49], their limitations come from the restricted deposition thicknesses and easy-delamination characteristic that will require redeposition after a long time of operation [26]. To overcome these disadvantages, attention has been paid to trigger the in situ formation of these carbon-based films via the tribological approach, which is based on tribochemical and/or tribocatalytic reactions [70,71] to replenish the carbon-rich layers from carbon-precursor sources [26]. In the pioneering work of Erdemir et al. [36], by coating the AISI52100 steel surface with copper-rich clusters through sputtering $MoN_x$-Cu

nanocomposite, the 10 h ball-on-disc tribo-experiment under 1.3 GPa normal pressure at room temperature results in a steady coefficient of friction (COF) of 0.08. This result is lower than the uncoated substrates in fully formulated oil (containing ZDDP) with a COF of about 0.1 (Figure 3a). In addition, the wear loss of the ball against the coated steel (with $7.35 \times 10^{-15}$ m$^3$) is also significantly lower than the ones against the uncoated steels (with $1.26 \times 10^{-15}$ m$^3$ and $1.34 \times 10^{-12}$ m$^3$ in the sliding tests with fully formulated oil and pure PAO 10 oil, respectively). Along with the mechanical enhancement of the MoN$_x$-Cu coating [9], the outstanding tribological performance of this coating was explained by the formation of carbon-based tribofilm from the PAO10 oil base. Its Raman spectrum was significantly similar to the graphitic DLC coating with a high sp$^2$ bonding portion. Moreover, this was also verified via the analysis of the resultant wear debris using electron energy loss spectroscopy (EELS), where graphite-like carbon nanocrystals (with 82% sp$^2$-portion) were detected. This study has paved a pathway to utilize active metallic atoms within hard coatings to promote the formation of carbon-based tribolayers and further advance their lubricating properties [72–74]. For example, using the same coating MoN$_x$ with Ag dopant, Xu et al. [73] demonstrated the ability to form onion-like carbon tribofilms from different hydrocarbon precursors. In this work, the rotation tribotests of 1-octadecene, octadecane, and oleic acid lubricating mediums have been carried out under 0.57 GPa Hertzian pressure at room temperature. According to the experimental outcomes (Figure 3b), the test of MoN-Ag coating with octadecane resulted in the best tribological performances, with just $0.62 \times 10^{-6}$ mm$^3$/Nm wear rate and 0.1 friction. The outstanding lubrication of this medium was later explained by its saturated structure, enabling strong adsorption on the active Ag cluster without the loss of amorphous carbon yield due to the CO$_2$ formation from the –COOH group.

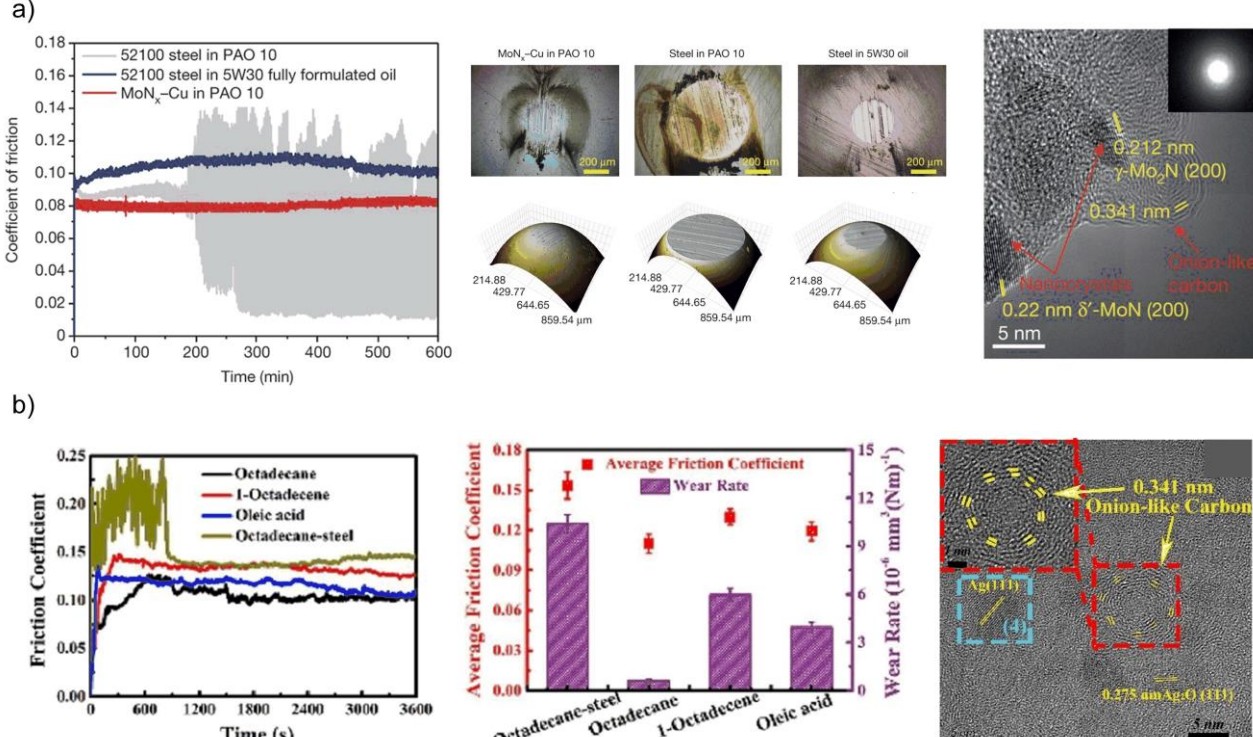

**Figure 3.** (**a**) Friction and wear performances of the lubricated tribo-pairs from the pioneering work of Edermir et al. [36]. Reprinted with permission and Copyright from Springer Nature 2016. (**b**) Friction and wear performances of the tribo-pairs lubricated by different organic additives in the inspirited study by Xu et al. [73]. Reprinted with permission and Copyright from Elsevier 2023.

Further works utilizing the same concepts of catalytic coatings were conducted with pure catalytically active alloy films. In the studies of Curry et al. [75], Pt-Au alloy film was introduced as a novel coating with an extraordinary antiwear property due to the thermally stable nanocrystalline of the alloying characteristics. The film $Pt_{0.9}Au_{0.1}$ of 2μm thickness deposited on 440PH stainless steel discs has a nanohardness of $7.1 \pm 0.4$ GPa. It slid against $Si_3N_4$ balls and sapphire balls as harder counterparts (with 15 and 25 GPa nanohardness, respectively) under 1.1 GPa maximum Hertzian pressure and 105 passes of 1 mm/s sliding speed at 20 °C. The maximum result wear rate of the disc was just $3 \times 10^{-9} \pm 10^{-9}$ mm$^3$/Nm, which can be classified as atomic attrition [76]. The resulting friction was relatively high (ranging between 0.25 and 0.3), which can be explained by "third body" sliding. It is concluded that the wear property depends on the bulk characteristic, while the friction performance is attributed to the shear layer. To further investigate the tribochemical potential of this coating, the authors conducted similar tribotests with sapphire balls in a nitrogen-rich chamber containing ambient organics and adventitious carbon [77] (Figure 4). In this work, the friction was surprisingly decreased to just about 0.01, where the formation of 50–200 nm thick carbon-based tribofilms, which the authors claimed as hydrogenated-DLC-liked tribofilms, mixing with Pt-Au nanoparticles, was found on the worn surface. The formation of the resultant carbon-based tribolayer was attributed to the catalytic effect of both Pt and Au atoms on the organic species [78–80]. More importantly, its frictional outcome agrees well with the antifriction performance of a-C:H film in dry $N_2$ sliding conditions [32]. In addition, the structural similarity of the resultant carbon-based tribolayer to the chemical vapour deposit (CVD) a-C:H coating (with 20% H content) was compared via Raman analysis. Combined with the work performed by Erdemir et al. [36], it is clear that the structure of catalytic carbon-based tribolayer could be tailored based on the employment of different active metals, hydrocarbon precursors, and/or sliding conditions [81]. As a result, the later study by Shirani et al. [82] reported the promotion of highly graphitic carbon-based tribofilm when the tribotest with Pt-Au alloy was performed in ethanol ambient. Meanwhile, further research by DelRio et al. [83] optimized the performance of this tribolayer by altering the Pt and Au contents within the target coatings.

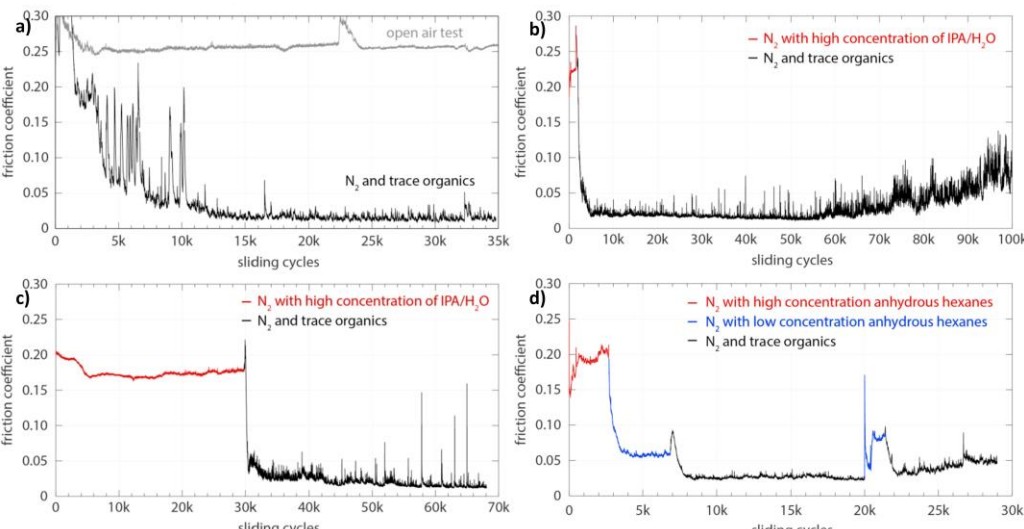

**Figure 4.** Frictional outcomes of $Pt_{0.9}Au_{0.1}$ alloy coating when the tribotests were carried out in lab air and inert gas comparison (**a**), inert gas with alcohol and water vapor (**b,c**), and inert gas with anhydrous hexanes (**d**) [77]. Reprinted with permission and Copyright from Elsevier 2018.

## 2.3. Carbon-Based Tribofilms Produced by Organic Additives

The employment of functional organic carbon-precursor additives, of which cycloalkane moieties are highly strained, has also been recognized as a distinctive way

to form the in situ carbon-based tribofilm between two sliding surfaces [37]. Compared to the hard catalytic coating approach, the deposition of these carbon-based tribolayers is much more convenient, since it does not require any pretreatment before the sliding process. Noticeably, in 2018, Johnson et al. [34] introduced cyclopropanecarboxylic acid (CPCa) as a promising sulphur-free and phosphorus-free additive for automotive engine oil. By dissolving 2.5 wt% CPCa in PAO4 oil, remarkable tribological performances can be clearly observed with an average 93% wear rate reduction and 18% friction reduction at room temperature, 0.05–0.2 m/s sliding speeds, and 10–20 N normal loads (Figure 5). From the Raman analysis of the CPCa samples' wear tracks, a similar configuration between the obtained carbon-based films and the commercial hydrogenated DLC coating was detected. Because the triboexperiments were carried out without any added catalysts, it is clear that CPCa performed as a carbon-precursor additive. Particularly, the cycloalkane fragmentation was confirmed to be triggered under high contact pressure and flash temperature, and eventually promoted the carbon-rich film. Further computational outcomes detected the polymerization process from CPCa fragmentation to promote hydrocarbons with high molecular weight [40]. These processes were later found to be associated with the strong absorption of the carboxyl group (–COOH) on the steel substrates [37,39,84].

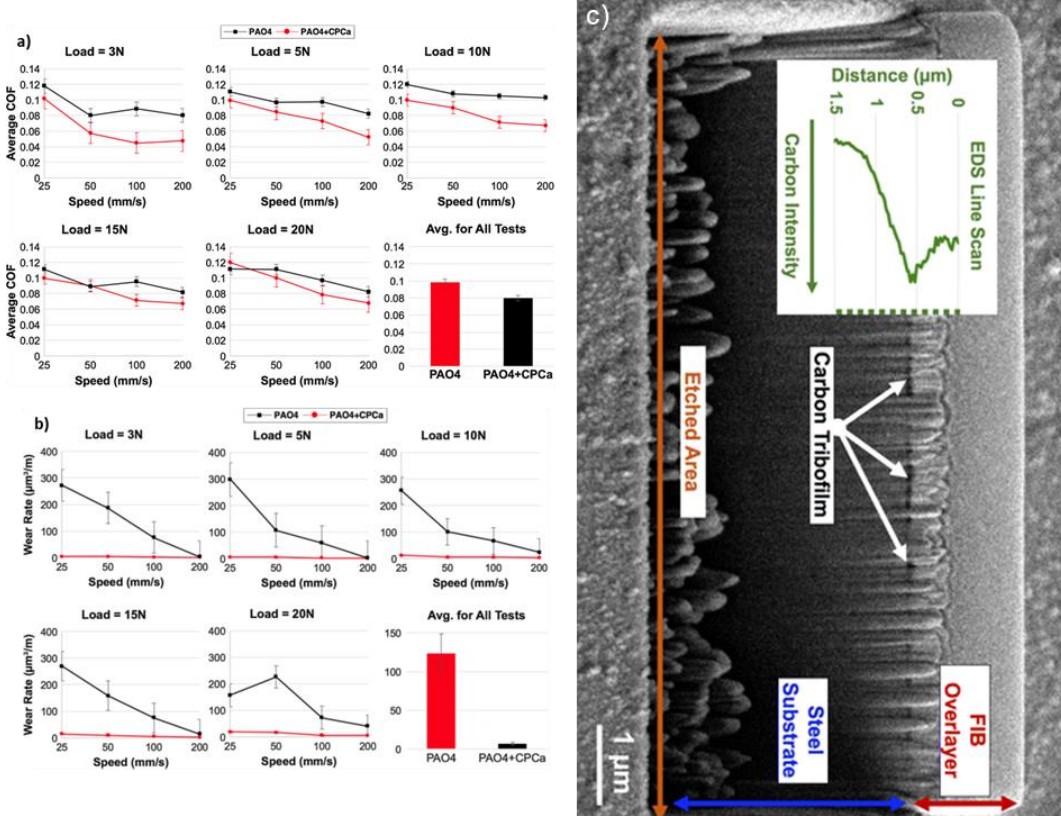

**Figure 5.** Friction (**a**) and wear (**b**) outcomes of PAO4 + CPCa compared to PAO4 under different sliding conditions at room temperature, and a FIB milling sample confirming the formation of dark carbonaceous tribofilm at the sliding surface of PAO4+CPCa tribotest (**c**) [34]. Reprinted with permission and Copyright from Springer Nature 2017.

Besides the cyclopropane moiety, such carbon-precursor additives with the cyclobutane group as cyclobutanecarboxylic acid (CBCa) are also found to promote polymeric carbon-based tribolayers under severe sliding conditions. However, according to Ma et al. [37], this moiety exhibits a much more stable characteristic, which dissociates and polymerizes much more slowly, resulting in inferior tribological performances compared to the cyclopropane one's tribotest under the same testing conditions (Figure 6a). Also, in

this work, cyclopropane-1,1-dicarboxylic acid (CPDCa) additive containing two carboxyl groups exhibited the best antiwear and antifriction properties in trimethylolpropane trioleate (TMPTO) base lubricant. By employing the molecular dynamics (MD) simulation, the superior performance of dicarboxylic acid additives was explained by their stronger binding strength on the sliding substrates than the monocarboxylic ones, resulting in negligible wear outcomes (Figure 6b). Finally, apart from the aforementioned additives, the polymeric carbon-based tribofilms activated from other organic molecules like *n*-butyl acrylate [85], *α*-pinene [86,87], allyl alcohol [47], etc., are also investigated prior to the practical applications.

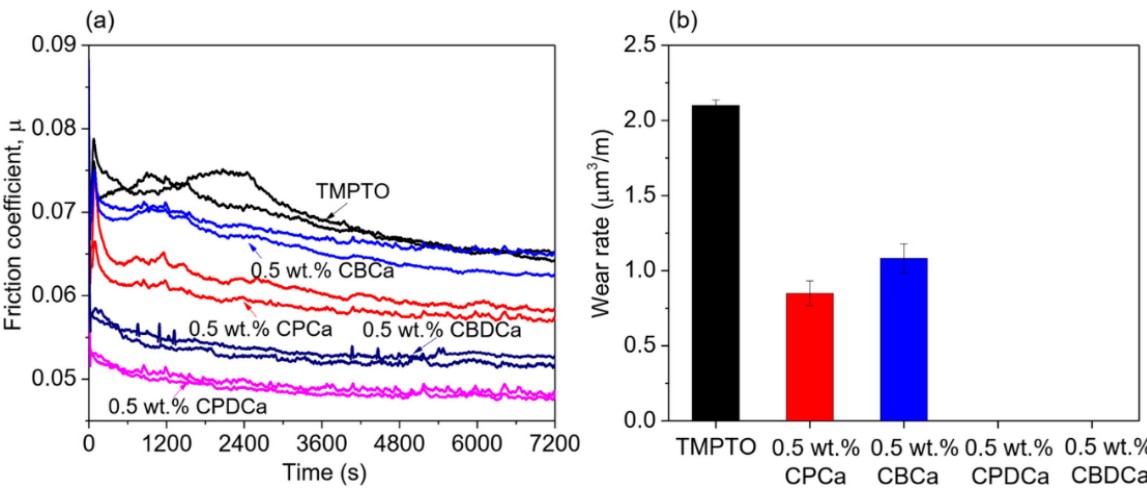

**Figure 6.** Friction (**a**) and wear (**b**) outcomes of different carbon-precursor additives in TMPTO under 10 N normal load and 0.05 m sliding speed at room temperature [37]. Reprinted with permission and Copyright from Springer Nature 2020.

Along with the functional additives, the chemical nature of the base lubricants can also exert dominant influences on the structures of in situ carbon-based tribolayers [88,89]. This has been highlighted in the current study by Rouhani et al. [35]. In his work, the four-ball tribo-experiments were carried out to evaluate the boundary lubrication performance of in situ carbon-based tribofilms generated from pure palm oil (PO), palm oil formulation (POF), mineral oil (MO), and mineral oil formulation (MOF), in which PO is Triacylglyceride (TAG)-based, which has received a lot of interest due to its ecologically friendly characteristics [90]. According to the four-ball tests (Figure 7), PO performs worse than MO in terms of seizure resistance at higher loads due to its reactive allylic protons. POF and MOF performed better; however, POF shows higher resistance to seizures than MOF at severe loading conditions. The authors believe that the antioxidants in the POF gave sufficient chemical protection to the TAG molecules, while surfactants enhanced the wettability of the steel, and viscosity modifiers optimized fluid properties. The Raman analysis of the resultant tribofilms from the POF oil compared to the one from MOF is demonstrated in Figure 7. Accordingly, at low experimental loadings, both POF and MOF produced carbon-based tribofilms that exhibited the resultant structure similar to a-C:H. Once the testing load increased, leading to an increase in seizure, the transformation from a-C:H to a-C, and eventually nanocrystalline-graphite-like structure (ncG), occurred [91], before the oil-based lubricants failed at the highest 400 kg load. The authors concluded from the Raman analysis that MOF oil is found to graphitize much easier than POF oil due to the appearance of aliphatic chains within the MOF oil base. These chains can promote kerogen formation [91–93], thus accelerating the graphitic transformation of resultant carbon-based tribofilms when the experimental load goes up. Combined with the corresponding tribological outcomes, the study concluded that the rapid graphitization of the polymeric carbon products can easily lead to early oil failure due to the failure in establishing the a-C tribofilms.

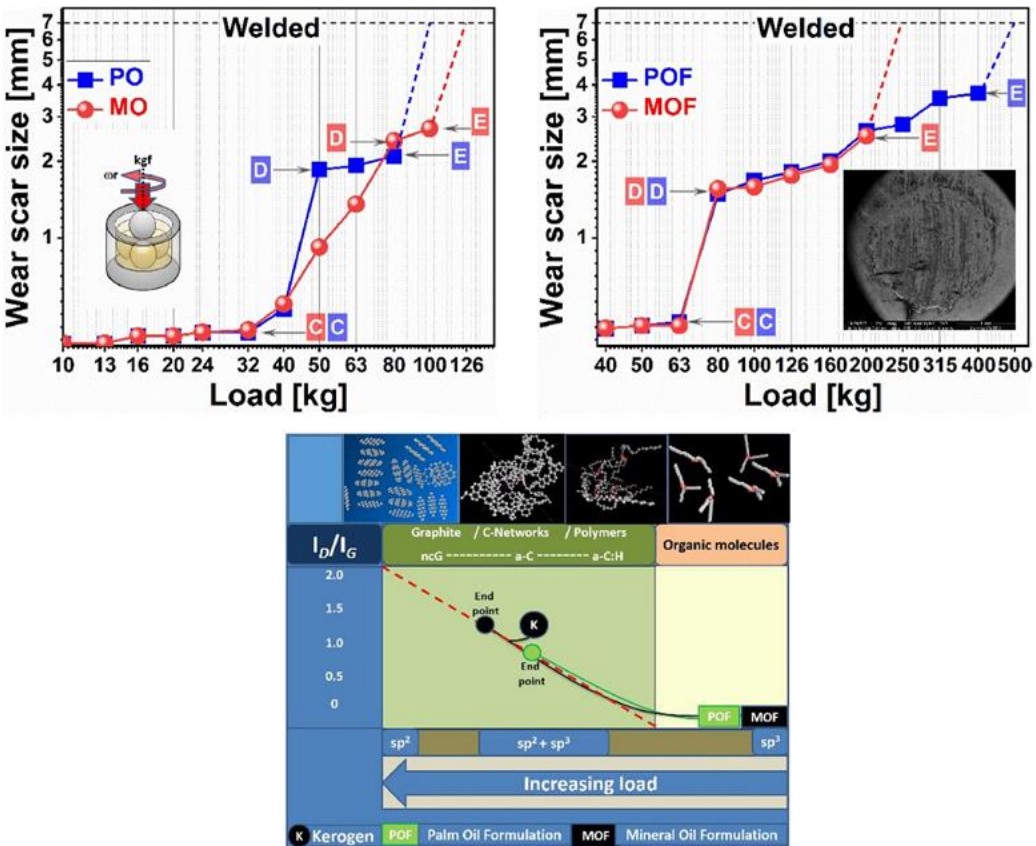

**Figure 7.** Average wear scar diameters resulted from 4-ball tests for different oil base lubricants and the Raman analysis showing the evolution of MOF and POF's $I_D/I_G$ ratio with a load [35]. Reprinted with permission and Copyright from Elsevier 2023.

### 2.4. Carbon-Based Tribofilms Produced by Catalyst Additives

Beyond the novel hard coating developments, interest in the catalyst agents has also spread to the lubricating nanomaterial investigation. For instance, Hu et al. [41] demonstrated the carbon-based tribofilm structure, which was formed via the catalytic effect of Ni nanoparticles under extreme-pressure conditions. It should be noted that nanoparticles have also been considered as another modern extreme pressure additive. They enter the microroughness to polish the interfacing surfaces during the sliding process, of which the four tribological mechanisms that allow for "smooth" contact from nanoparticles in oil lubrication are rolling, protective film, mending, and polishing effects [94,95]. Among them, the nickel nanoparticle emerges as one of the softest nanoparticles, with the lowest melting temperature, which is more beneficial for promoting a compact protective film on the contacting area via the nanoparticles' sintering by flash temperature [96]. In this example, cubic Ni nanoparticles with a 21.3 nm average size were synthesized from nickel formate and oleylamine in order to enhance their dispersion in the PAO6 oil base. Their test concentration in base oil was 1 wt%. The ball-on-disc tribotests were carried with 0.05 m/s sliding speed and 1800 s duration under the 100 N normal load. The experimental disc was made of 304 stainless steel, while the ball's material was GCr15 steel. From the experimental results (Figure 8), cubic Ni nanoparticles exhibited a remarkable tribological performance under extreme-pressure conditions. By employing the nanoadditive, the friction was dramatically reduced from 0.4 to around 0.1 (Figure 8a). Meanwhile, the average disc wear rate of the PAO oil containing Ni nanoparticles was just $0.89 \times 10^{-6}$ mm$^3$/Nm, which was ~92% lower than the pure oil test ($11 \times 10^{-6}$ mm$^3$/Nm) ((Figure 8b). Further analysis of the Ni test at 100 N demonstrated a compact film formation on the wear tracks that contained a high amount of nickel, carbon, and oxygen elements.

This indicated that during the sliding process, Ni nanoparticles had entered the interfacing area. Then, these particles oxidized partially and created an iron–nickel oxide layer with high binding energy. After that, the embedded Ni nanocrystals further promoted the carbonizing process of based oil to create a DLC-based layer with a high graphitization degree, resulting in low-friction outcomes (Figure 8c).

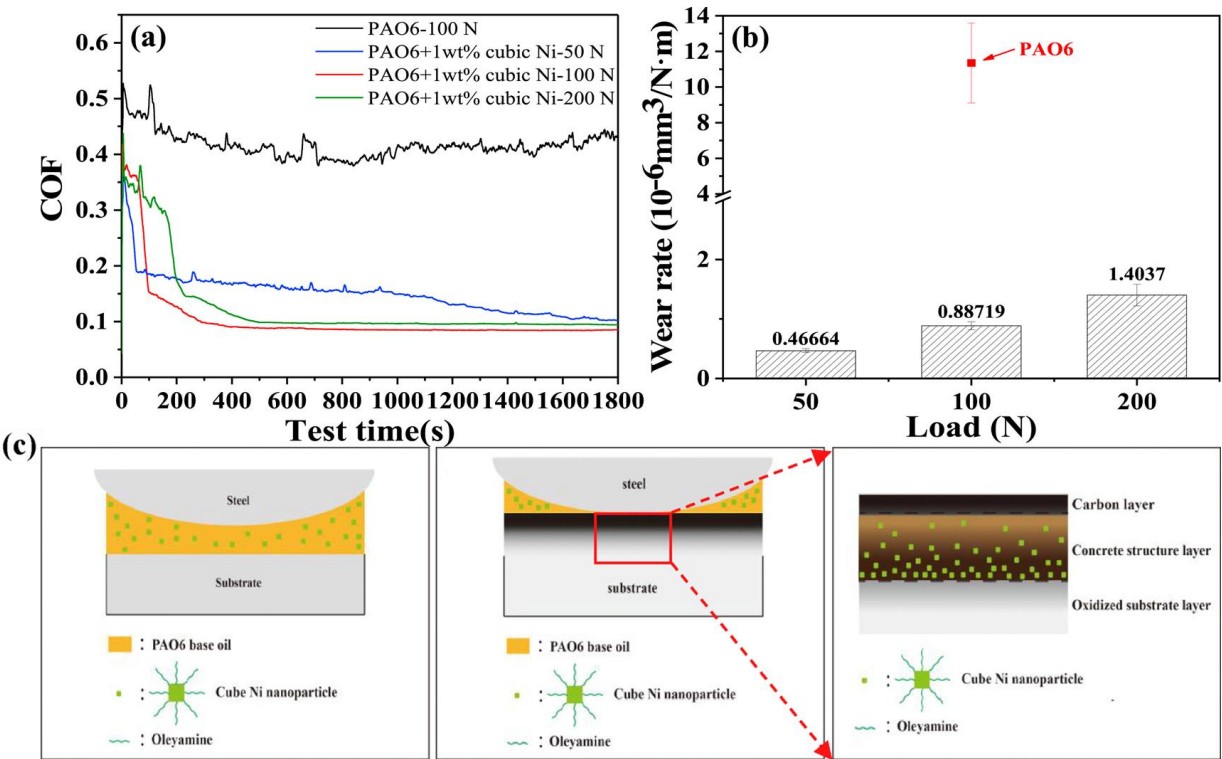

**Figure 8.** Tribotest outcomes (**a**,**b**) and lubricating mechanism (**c**) of 1 wt% Ni nanoparticle under extreme-pressure sliding conditions [41]. Reprinted with permission and Copyright from Elsevier 2020.

On the other hand, the concept of utilizing active elements in promoting and/or controlling in situ carbon-based material has also been applied in current research of 2D nanomaterials like layered double hydroxides (LDHs). It should be noted that LDH has already emerged as a "green" additive [97] that can achieve superlubricity performance in lubricant research [98] thanks to their manageable structures [99,100] as well as the outstanding shearing capability [101,102]. These nanomaterials are also famous for their catalytic applications, where many of which are well known as functional catalytic agents to promote the formation of helical carbon nanotubes, carbon nanofibres, and CNT/graphene hybrids from carbon-precursor gases [103–106]. Therefore, by utilizing both lubrication and catalytic functions of NiAl-CO$_3$ LDH [107,108], we have successfully advanced the tribological performances of CPCa under severe sliding conditions at different temperatures [109]. In this work, extreme boundary rotation tribotests, of which the lambda value was just 0.09, have been carried out under 25 °C, 50 °C and 100 °C. It is clear from the tribotest outcomes (Figure 9) that the combination of 2.5 wt% CPCa and 0.1 wt% NiAl-CO3 LDH in PAO4 ambient (CPCa + 0.1%LDH) results in the best tribological performances at all experimental temperatures. At room-temperature tests, the superior lubricating property of CPCa + 0.1%LDH was dominantly attributed to the easy exfoliation of the 2D nanomaterial's layered structure [110]. Once the testing temperature increased, the formation of hierarchical tribofilms was observed, where the Ni-based phases within protective interlayers from LDH were found to exert the catalytic effect, replenishing the upper carbon-based tribolayers. These unique multilayer tribofilm promotions not only stabi-

lized the performance of soft and low-absorption-strength carbon-based tribolayers from CPCa under high-temperature tribotests, but also exhibited better antiwear and antifriction properties than the conventional ZDDP tribofilm under similar sliding conditions.

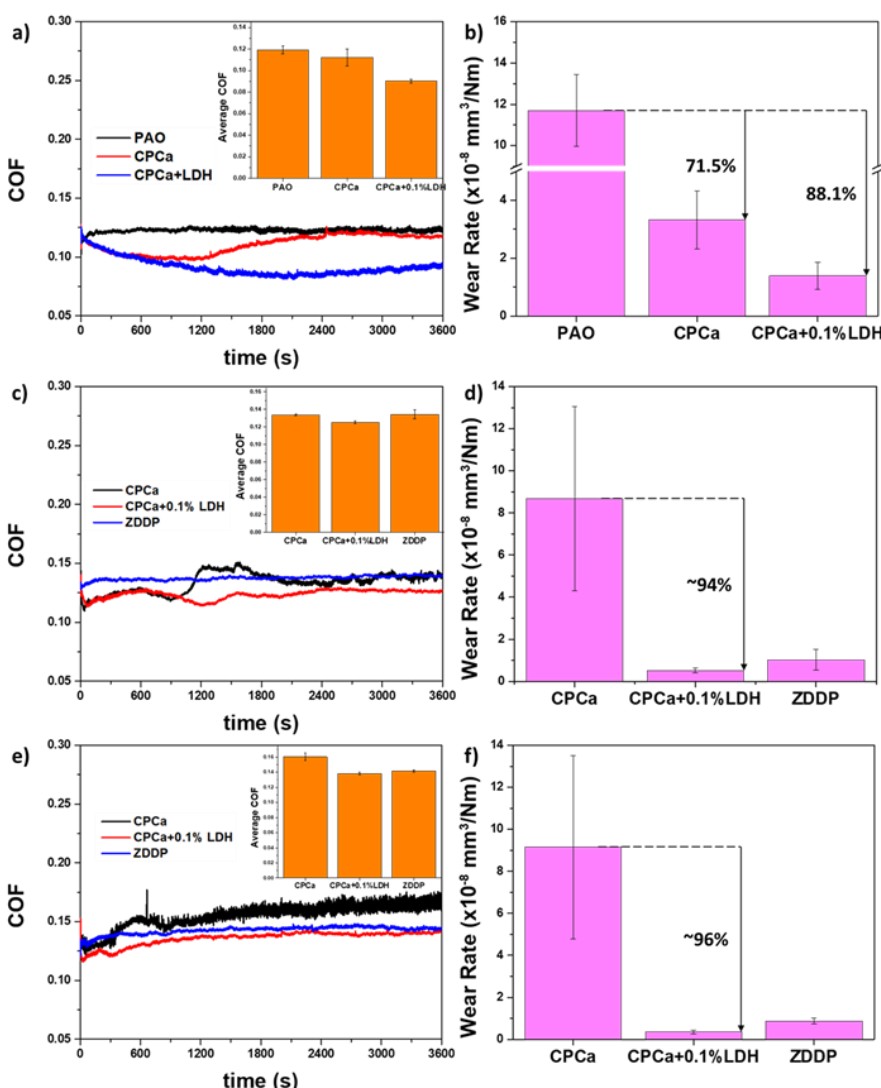

**Figure 9.** Tribological performances of PAO4 oil sample containing 2.5 wt% CPCa and 0.1 wt% NiAl-CO3 LDH (CPCa + 0.1%LDH) compared to other benchmark lubricants under 100 mm/s sliding speed and 10 N normal at 25 °C (**a,b**), 50 °C (**c,d**), and 100 °C (**e,f**) [109].

## 3. Tribochemistry and Mechanistic of the Formation of Carbon-Based Tribofilms

Based on the previous section, the contribution of flash temperature and the shear stress at rubbing contacts have been highlighted as the key factors for the formation nature of in situ carbon-based tribofilms. According to the recent review of Berman et al. [26], the mechanism underlying the carbon tribofilms' formation is the tribocatalysis process induced by the catalytic metal surface. There are two major steps involved in the formation of carbon tribofilms: (i) the dissociation of the C–H bonding and the scissoring of the C–C backbone to form active radical carbon fragments (Figure 10a); and (ii) the polymerization and reconstruction of the fragments into solid carbon films (Figure 10b). In the very recent study of Rouhani et al. [35], Raman analysis was used to track the chemical structure of the thermal products from ancient oils on a wear track at different temperatures, which qualitatively confirms these two steps' formation of the carbon tribofilms. Qi et al. [80] used density functional theory simulation to suggest that the formation of the carbon tribofilms starts near the active metal surface where the energy barrier for the reaction step (ii) is

reduced. Meanwhile, cyclic deformations due to subsequent shear stress produce energy accumulation at metal surfaces, which also accounts for the scissoring and dissociation of the hydrocarbon precursors during sliding in step (i). According to Wu et al. [40], the rate-limiting step in carbon tribofilm formation is the fragmentation of hydrocarbon precursors (e.g., CPCa), thus highlighting the important role of energy generated at the sliding interface.

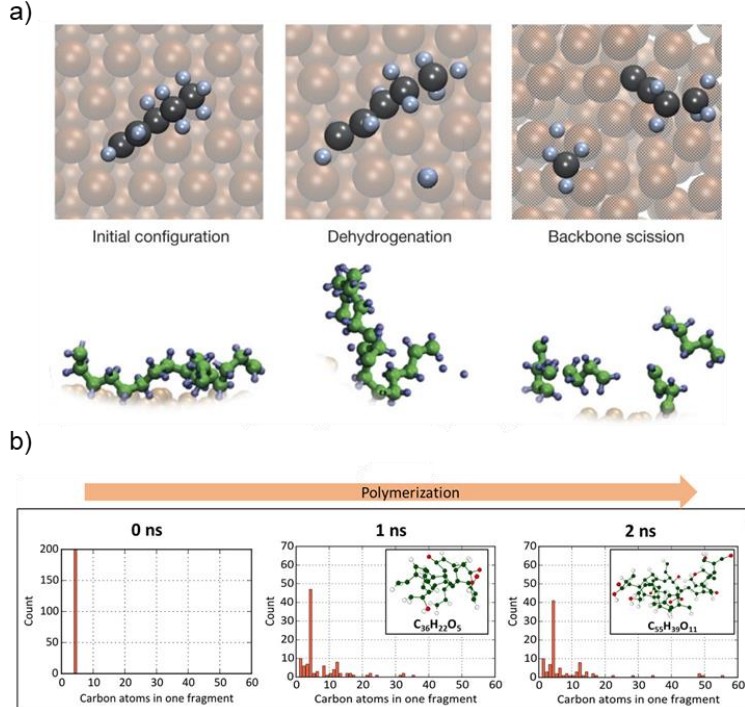

**Figure 10.** Tribocatalysis mechanisms on the formation of the in situ carbon-based tribofilms from ref [36] (**a**) and [40] (**b**). Reprinted with permission and copyright from Springer Nature 2016 and American Chemical Society 2019, respectively.

Various studies have suggested several factors driving the increase in energy at the rubbing surfaces, including flash temperature rise, pressure, triboemission, and shear-stress-promoted thermal rise. Among these, flash temperature rise is the most common factor believed by various researchers in the field. Flash temperature is defined as heat generated when solid surfaces are rubbed together, which causes a local and transient temperature rise at the contacting interface. Jaeger et al. [111] developed the moving heat source theory to calculate the magnitude of this temperature rise, which shows a strong dependence of flash temperature rise on the rate of heat generation, the speed of the moving surfaces with respect to the steady contact, the thermal properties of the solid surfaces, and the contact dimensions. The estimated flash temperature rise at the interface between the ball and the disc for the Hertzian circular contact can be derived as:

$$\theta = \frac{\dot{Q}_{in}}{r.K} \frac{0.5895}{\sqrt{J + 2.471}} \tag{1}$$

where K is the thermal conductivity, $\dot{Q}_{in}$ is the heat dissipation rate or total rate of heat generation, J is a dimensionless speed of the particular surface relative to the contact, and r can be treated as the Hertzian contact radius. According to the theory, the flash temperature rise is substantial at high sliding speeds and can be used to explain the formation of tribofilms with fast-moving contact systems. However, it fails to explain the formation of the tribofilms at sliding speeds below 0.1 m/s where the boundary lubrication conditions are often tested. In our current work [109], the estimated flash temperature rises at different

testing temperatures are small, and they cannot be used solely to explain the formation of the continuous carbon tribofilms as well as tribo-oxide layers. It should be noted that the theory used to calculate flash temperature rises is based on the assumption of a Boltzmann energy distribution at the macroscale, and it may not be true when accounting for the very intense energy dissipation at individual asperity conjunctions. Indeed, the molecular dynamic simulation conducted by Wu et al. [40] suggested an interfacial temperature rise from 300 to 730 K at individual asperity contacts, which is sufficiently high to fragment the hydrocarbon precursors, step (i); and trigger the polymerization and reconstruction of the solid carbon films (ii).

In the theoretical work of Kajdas et al. [112], a new concept on the mechanism of tribo-catalytic reactions was introduced, taking into account the important role of mechanical forces on the chemical reactivity of the molecules. According to the authors, mechanical energy is transformed into a flux of electrons/photons, termed as triboemission, in the tribocatalysis process, which provides additional amounts of energy to the lubricating systems. This triboemission may be responsible for tribochemical reactions between contacting solids and lubricant molecules which can explain the formation of the carbon-based tribofilms. The relationship between the modified coefficient of reagent (i.e., hydrocarbon molecules in lubricant) reactivity $\alpha_i$ and the mechanical energy acting on a tribological system can be expressed as:

$$\alpha i = \frac{L - L_0}{Ae^{-\frac{E_a}{RT+\varepsilon}}}[(e_0\cos(k_2L + k_3)]t \tag{2}$$

where $(L - L_0)$ describes a difference between the mechanical work performed, necessary to reach the critical condition, between the observed tribological system and the reference tribological system. Both $L$ and $L_0$ are a function of the applied load, sliding speed, and rubbing time (t). $T$ is the temperature of the boundary lubricant layer; $\varepsilon$ is the energy taken in any form other than heat introduced into the reaction space; $e_0$ is a flux of energy emitted by the surface of the solid as electrons or photons; $k_2$, $k_3$ and $A$ are the constant values, where $k_2$ and $k_3$ depend on the surface properties of the solid contacting elements. It can be noticed from Equation (2) that part of the reactivity of the molecules depends on the energy generated from the mechanical work performed, $L$, which is transported from the external part of the solid elements to the contacting surfaces, i.e., the reaction systems. This mechanical energy is responsible for a change (i.e., an increase) in the internal energy of the reaction media which is expressed as $C$ and is a multiplication between a ratio of the trigonometric part and Arrhenius part and the time (t) of equation (2). It is believed that the change in the internal energy is distributed into a liquid/fluid phase, which leads to an increase in temperature in the lubricant. Meanwhile, the energy ($\varepsilon$) accumulated at solid surfaces undergoes triboemission, in the form of electron or photon flux, to the triboreaction space. These emissions can be absorbed by lubricants/additive molecules, providing sufficient energy to the molecules to overcome the reaction barriers $E_a$. Thus, conclusions in the new theory of Kajdas et al. [112] can be used to explain the ability of the catalysis reactions in the tribological systems to proceed at temperatures much lower than for static contact or traditional catalytic systems.

Apart from triboemission, additional energy responsible for tribochemical/tribocatalytic reactions at the sliding interfaces may also come from the shear stress presented during rubbing. The concept of physical and chemical reactions induced by applied shear stress has been proposed since the last century, and the most used models to relate the chemical reactions and the mechanical forces are the Eyring model [113] and the stress-assisted thermal activation model of Arrhenius [114]. These models, in particular, are the foundations of the development of the emerging field of mechanochemistry, which is devoted to studying the influence of applied shear stress on the chemical reactions between different materials. In tribological systems, mechanical forces are always presented in the form of shear stresses; thus, these models are particularly relevant to explain the tribochemical reaction kinetics. While the Eyring model uses a thermally activated cage model to explain the tribofilm

formation from liquid lubricants, this model is limited in quantitatively analysing the growth rate of tribofilms. On the other hand, the Arrhenius model provides a detailed description of the influence of temperature and applied shear stress on the probability of molecules undergoing physical and chemical processes, which can be further scaled up into the rate constant of the tribofilm formation [114]. The simplest expression of the Arrhenius model that describes the growth rate, k, is presented as:

$$k = k_0 \, e^{-\frac{E_0 - \sigma \Delta v}{k_B T}} \tag{3}$$

where $k_0$ is the pre-exponential factor, $E_0$ is the activation energy or energy barrier for chemical processes without applied shear stress, $\sigma$ is the mean value of the applied shear stress, $\Delta v$ is the activation volume, $k_B$ is the Boltzmann constant, and T is the absolute temperature. According to Equation (3), the growth rate of the tribofilms is increased due to the applied shear stress by a factor of $e^{\sigma \Delta v / k_B T}$. On the other words, the applied shear stress reduces the activation energy or energy barrier for chemical processes to occur. Figure 11a clearly depicts the influence of applied force (i.e., shear stress) on the activation energy profile of a chemical reaction [115]. The solid line represents the energy path that the chemical reaction follows over the potential energy plane from reactant to product with no applied shear stress, where R is the reactant, TS is the transition state, and P is the product configuration. Meanwhile, the dashed line shows how the energy profile plummets with the applied mechanical force (i.e., shear stress). This model has been successfully used to study the growth rate and kinetics of the ZDDP tribofilm formation under boundary [116] (Figure 11b) and elastohydrodynamic (EHD) lubricated conditions [117]. Recently, in the study by Johnson et al. [34], this model was used successfully to qualitatively construct the activation energy plot for carbon tribofilm formation from CPCa versus shear stress (Figure 11c), with an assumption that the carbon friction products accumulate inside roughness grooves of the surfaces during contact, while friction is continually reduced until reaching a steady-state value, where the roughness grooves are filled with the carbon tribofilms.

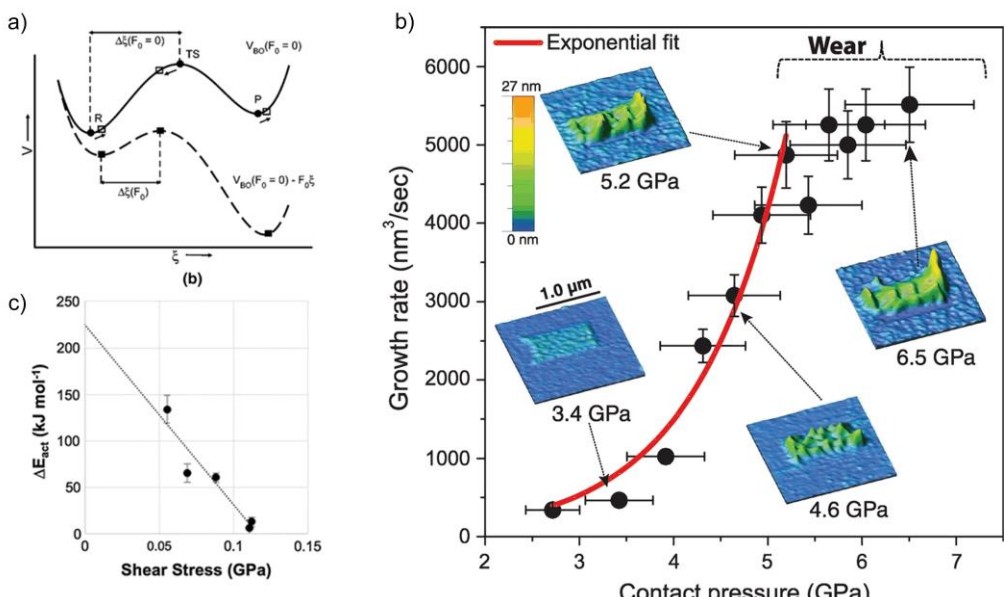

**Figure 11.** (**a**) Diagram showing the influence of applied force on activation energy in a chemical reaction [115]. (**b**) Tribofilm volumetric growth rate dependence on contact pressure, as studied by AFM with ZDDP tribofilms [116]. (**c**) Calculated activation energy of carbon-based tribofilm formation from CPCa versus shear stress [34]. Reprinted with permission and Copyright from American Chemical Society 2012, Science 2015, and Springer Nature 2017, respectively.

## 4. Characterizations of Carbon-Based Tribofilms by Analytical Techniques

Along with the recent breakthrough achievements in carbon-based tribofilm investigation, the suitability of the analytical technique is also highly focused to solely characterize the nature of the resultant tribolayer. This section will provide basic information of the most common analytical tools and their employments in this research field.

### 4.1. Raman Spectroscopy

Raman is considered a useful technique that has been widely utilized in the majority of studies requiring the characterization of the carbon materials' bonding states [65]. The popularity of this technique comes from its nondestructive characteristics and availability, which allows for the direct analysis of the experimented surfaces [69]. The common carbon materials' Raman signals are illustrated in Figure 12a, in which the G band (~1560 cm$^{-1}$) is associated with the stretching of the carbon signal with the sp$^2$ bond in the structure of crystalline graphite, while the D band (~1360 cm$^{-1}$) corresponds to the edges' vibrations, which are associated with the noncrystalline and/or the disordered characteristic within the carbon matrix [30,118]. It should be noted that the Raman spectra of the carbon-based materials with higher crystal degrees will appear with the sharper D peak and G peak [119]. Furthermore, the obtained I$_D$/I$_G$ fraction from Raman outcomes has been evaluated as useful data to estimate the carbon matrix's sp$^2$/sp$^3$ ratio, where I$_D$ and I$_G$ are amplitudes of D and G peaks, respectively [69].

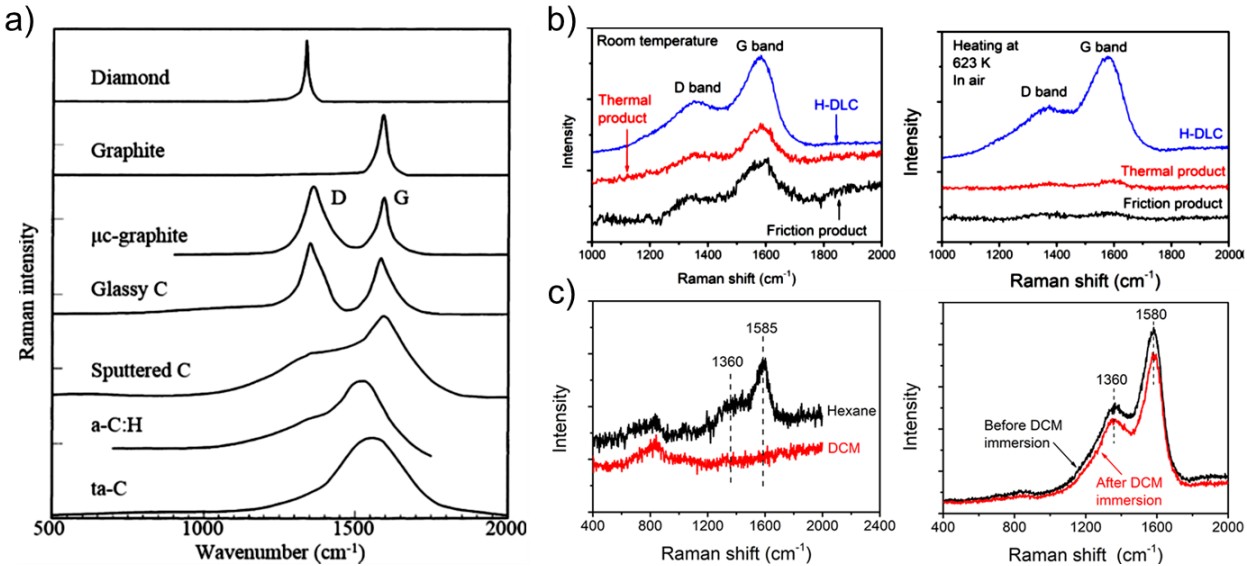

**Figure 12.** (**a**) Typical Raman signal of the common carbon materials [53]. (**b**) Raman signals of CPCa's experimental products compared to H-DLC at room temperature and 350$^0$C and (**c**) Raman signals of friction product after being rinsed by hexane and CH$_2$Cl$_2$ DCM and H-DLC layer before and after immersion in DCM for 20 h [40]. Reprinted with permission and Copyright from Elsevier 2002, and American Chemical Society 2019, respectively.

Despite its convenience, the use of the Raman technique alone is not the sole approach to fully characterize the nature of carbon-based tribofilm. This had been indicated in the CPCa's tribofilm-forming mechanism study by Wu et al. [40]. In this work, a similar triboexperimental material and experimental duration (30 min) were carried out at 10 N and 0.2 m/s sliding speed under the boundary lubrication regime. As expected, CPCa effectively improved the tribological performance of PAO oil. The formed dark-coloured deposit as the frictional product was collected by cleaning the wear track with hexane. Its measured thickness was around 2.60 $\pm$ 0.08 μm. Meanwhile, the thermal deposition's carbon material was produced by mixing 0.3 g CPCa with 0.1 g Fe$_3$O$_4$ as the typical oxide phase of the steel surface [120] at 200 °C for three hours. Moreover, H-DLC as standard

DLC film with a 40% hydrogen portion, which was deposited on the silicon substrate via chemical vapour deposition from a gas mixture of 25% methane and 75% hydrogen, was also compared with the friction and thermal products. After that, Dichloromethane ($CH_2Cl_2$) was used to dissolve the obtained residual product before being filtered and dried. According to the Raman analysis (Figure 12b), the thermal product and friction product produced the same Raman peaks as H-DLC at room temperature. However, their resultant peaks had been removed when the temperature was elevated to 350 °C.

Moreover, the friction product was detected to be dissolved by $CH_2Cl_2$ (DCM), while the H-DLC's Raman signal remained intact after 20 h immersion in $CH_2Cl_2$ solution. Therefore, it is clear that the in situ carbon-based tribofilm from CPCa had a similar configuration to the thermal product, and they are not similar to DLC film. Further computational outcomes using reactive force field (ReaxFF) and MD simulations detected the polymerization process from CPCa fragmentation to promote hydrocarbons with high molecular weight. As a result, it can be concluded that the Raman analysis alone may not be enough to conclude the carbon-based tribofilm nature and/or structure, especially the one from the oil-soluble additive where the oil decomposed product and/or surface's oxide particles can exert certain effects on the obtained spectra [121,122]. Similar concerns on characterizing carbon-based tribofilms using Raman were also raised in a very recent study [123], in which experimental evidence demonstrates that the D- and G-band features in Raman spectra of carbon-based tribofilms may originate from the photochemical degradation of carbonaceous organic matters induced by the Raman laser. Thus, the observation of the D- and G-band features in Raman spectra is not sufficient to confirm that DLC, a-C, or graphitic species are produced via tribochemical reactions assisted by interfacial friction or shear of organic molecules.

*4.2. X-ray Photoelectron Spectroscopy (XPS)*

Besides Raman spectroscopy, the use of X-ray photoelectron spectroscopy (XPS) as a surface-sensitive analysis is also very common in characterizing carbon-based films. The function of this technique is to qualify the elemental states as well as their chemical composition, using the photoelectric effect to define the elemental formation and their bonding network. Therefore, compared to Raman, the XPS technique can provide more details relating to the bonding configuration, because it indicates the specific carbon atom's chemical bonding in the C 1 s core-level spectrum [124]. Accordingly, the $sp^2/sp^3$ ratio of the carbon-rich films could be obtained via such deconvoluted peaks as C–C $sp^3$ and C–C $sp^2$, centring at 285.2 eV and 284.3 eV binding energy, respectively [125,126]. This technique has been used predominantly to characterize carbon-based films, assuming that the carbon-based tribofilms have chemical compositions like DLC films. From that, the hydrogen content of the carbon-based films can be estimated via the following equation:

$$sp^2/sp^3 = \frac{8 - 13[H]}{6[H] - 1} \qquad (4)$$

where [H] denotes the total hydrogen content of the coatings/films. The correlation between the $I_D/I_G$ fraction of Raman outcomes and the $sp^2/sp^3$ ratio collected from the XPS technique has been demonstrated in the work performed by Wei et al. [66]. In this study, the influence of Al and Cr as two metallic interlayers on the sputtered DLC films via the plasma-enhanced chemical vapour deposition (PECVD) approach on the SUS304 steel and silicon substrates was investigated. Two different interlayer thicknesses, namely, 20 nm and 40 nm, were analysed. Detailed calculated thermal stress via the finite element analysis of all DLC films can be found in Table 1. Moreover, the bonding structure of these resultant DLC films was also demonstrated. Particularly, the $I_D/I_G$ was collected by using Raman spectroscopy outcomes. XPS was employed to measure the hydrogen, $sp^2$, and $sp^3$ contents in the DLC layers. In short, the hydrogen and $sp^3$ contents inside the DLC films on steel substrates increase with an increase in metallic interlayer thickness. In contrast, these contents inside DLC layers on silicon substrates decrease with the higher metallic

interlayer thickness. It should be noted that the higher the $sp^3$ content, the higher the hardness of DLC film would be [50]. In addition, the increase in $sp^3$ content also results in the increase in $I_D/I_G$ ratio, indicating lower graphitic carbon-rich layer formation with lower $sp^2$ content. As a result, it is considered to employ the XPS technique to detect the exact bonding ingredient within the carbon-rich matrix.

**Table 1.** Obtained bonding data of all resultant DLC layers [66]. Reprinted with permission and Copyright from Elsevier 2009.

| Coating Systems | $I_D/I_G$ | $sp^2/sp^3$ | $sp^3/(sp^2+sp^3)$ (%) | $[H]_{total}$ (%) | Thermal Stress (MPa) |
|---|---|---|---|---|---|
| DLC-Si | 0.712 | 0.95 | 51.2 | 48.0 | 16.87 |
| DLC-(20 nm Cr)-Si | 0.669 | 1.64 | 37.9 | 42.2 | 36.50 |
| DLC-(20 nm Al)-Si | 0.673 | 2.01 | 33.2 | 40.0 | 55.30 |
| DLC-(40 nm Cr)-Si | 0.664 | 2.17 | 31.5 | 39.1 | 48.64 |
| DLC-(40 nm Al)-Si | 0.296 | 2.65 | 27.4 | 36.9 | 75.95 |
| DLC-Steel | 0.939 | 1.07 | 48.3 | 46.7 | 179.70 |
| DLC-(20 nm Cr)-Steel | 1.553 | 0.50 | 66.6 | 53.1 | 178.10 |
| DLC-(20 nm Al)-Steel | 1.361 | 0.80 | 55.6 | 49.5 | 115.00 |
| DLC-(40 nm Cr)-Steel | 1.488 | 0.48 | 67.6 | 56.0 | 177.10 |
| DLC-(40 nm Al)-Steel | 1.604 | 0.64 | 61.0 | 51.3 | 111.30 |

Also, the study by Wei et al. [66] successfully highlighted the usefulness of metallic interlayers in advancing the adhesion of carbon-based coating on the steel substrate. From the collected atomic force microscopy (AFM) outcomes, describing the morphology of DLC films under peeling conditions, these metallic layers have successfully enhanced the DLC film's adhesion to the steel substrates. In contrast, both Cr and Al interlayers failed to support the DLC film on the silicon bases. This discrepancy can be explained by the difference in the coefficient of thermal expansion (CTE) between the DLC and the silicon substrates [64]. The metallic interlayers would have a beneficial effect on the DLC film's adhesion with a small CTE deviation between the DLC film and the steel surface, resulting in lower thermal stress. Meanwhile, in the case of a high CTE difference, like the one with the silicon surface, the introduction of these metallic layers would increase the thermal stress that negatively affected the absorption of the DLC layer.

*4.3. Scanning/Transmission Electron Microscope (S/TEM) and Electron Energy Loss Spectroscopy (EELS) Combination*

Along with the analysis tools like Raman and XPS, electron energy loss spectroscopy (EELS) has also been recognized as one of the most typical techniques that can be employed to analyse the chemical structure of the resultant tribofilm, especially carbon-based tribofilm. This technique obtains the energy loss of the electron that undergoes the inelastic scattering, including plasmon excitations, Cherenkov radiation, inner shell ionizations, intra- and intertransitions, and phonon excitations, in which, it is useful to detect a material's elemental components via inner-shell ionizations. For instance, the needed energy amount for an inner-shell electron removal of the carbon atom is about 285 eV. If the collected energy is 285 eV lower than the input one, the appearance of carbon in the specimen can be confirmed. Moreover, based on the obtained energy loss range, the atom types will also be determined. When EELS is combined with TEM, it can provide information relating to the chemical or structural characteristics of a solid at the atomic scale [127]. This combination has been considered a useful approach to analyse the resultant tribofilm with the hierarchical structure.

A typical instance of how the EELS technique can be employed to analyse the carbon-based layers in the hierarchical tribofilm is the work conducted by Pham et al. [119]. In this work, EELS analysis was conducted on the tribofilms at identified areas to obtain the

orientation of graphite crystal in the resultant tribofilm structure. The integration window approach was applied to obtain the $sp^2$ hybridization semiquantification, in which the C-K edge spectrum's $\frac{I_{\pi*}}{I_{\pi*}+I_{\sigma*}}$ ratio was determined by employing the window energy of 15 eV and 8 eV for the $I_{\pi*}+I_{\sigma*}$ and $I_{\pi*}$ features, respectively. From that, the fraction of $sp^2$-C can be estimated via the following equation:

$$\frac{sp^2}{sp^2+sp^3} = \frac{\frac{I^S_{\pi*}}{I^S_{\pi*}+I^S_{\sigma*}}}{\frac{I^r_{\pi*}}{I^r_{\pi*}+I^r_{\sigma*}}} \tag{5}$$

where $I^S$ is the integrated intensities of the resultant tribofilm, while $I^r$ represents the reference value of pyrolytic graphite with a 100% $sp^2$ bonding fraction. The calculated $\frac{I^r_{\pi*}}{I^r_{\pi*}+I^r_{\sigma*}}$ ratio as the reference value based on the EELS spectrum of pyrolytic graphite was 0.288. In the case of the resultant tribofilm from the triboexperiment with the graphene-free oil sample, the determined $sp^2$ fraction in the carbon matrix was just 32%. This indicated the formation of the amorphous carbon-rich layer with a dominant $sp^3$ fraction originating from oil decomposition within the polyphosphate matrix of the ZDDP film [128]. Meanwhile, the tribotests with GNPs resulted in tribofilms with hierarchical structures. It is clear from their collected STEM-EELS analysed images (Figure 13a) that the $sp^2$ fraction of the resultant carbon-rich layers varied within their thickness. It should be noted that these collected spectrums are always the intermediate value between the added graphene and the amorphous carbon from the oil base. Due to the appearance of GNP, the $sp^2$ fraction of these layers was more than double compared to the graphene-free sample. Moreover, the significant high $sp^2$ fraction that can be detected near the steel substrates is attributed to the mechanical reinforcement effects of added graphene nanoplates (GNP). They have been exfoliated or deformed into fewer-layered graphene and compacted into the oxide layer of the steel substrates, increasing the interfacing areas' hardness (as verified by the supporting nanoindentation tests). In the case of the oil sample with 0.5 wt% graphene (Figure 13b), the formation of the large graphene clusters within the carbon-based films near the steel substrates can be further verified alongside the low-thickness ZDDP film promotion. Such clusters can be easily identified due to their parallel orientation to the beam, resulting in strong scattering.

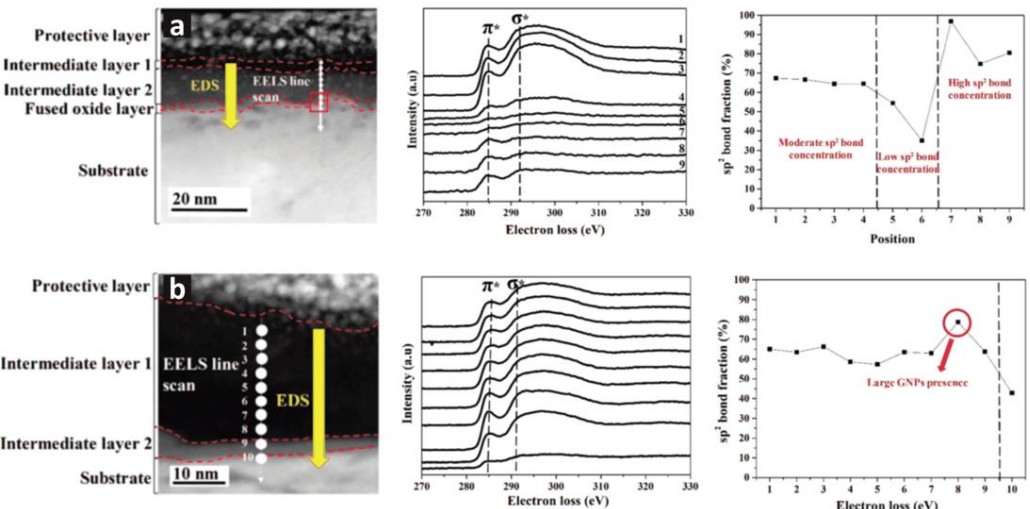

**Figure 13.** STEM-EELS line analysis of the resulted tribofilms from the tribotest with formulated oil containing (**a**) 0.05 wt% GNP and (**b**) 0.5 wt% GNP [119]. Reprinted with permission and Copyright from John Wiley and Sons 2019.

Another noticeable example that utilizes the EELS technique to distinguish different types of carbon-based nanostructure is the work by Ramirez et al. [129]. In this work, the Ni-VN coating (deposited on AISI 52100 steel substrate) was applied to promote the formation of carbon-based tribofilm with numerous nanostructures from methane gas ($CH_4$) as the carbon-rich fluid. From the experimental outcomes, the coated contacting surfaces exhibited a better tribological performance compared to the uncoated ones with around 50% lower friction and more than two orders of wear reduction. The superior lubricity of this coating can be explained by its ability to promote the tribochemical reaction that dissociatively extracts the methane gas to create carbon-based tribofilm on the Interfacing surfaces [36,130]. The generated blackish wear particles around the contacting areas were also collected and analysed using the EELS technique combined with HR-TEM. It is clear from the characterized outcomes that there were main types of wear particles, including the hollow-centred carbon onion with 5–10 nm diameter mixing with the 1- to 10-layered graphene (Figure 14B,C), and disordered graphitic structures wrapping agglomerated debris (Figure 14D,E). Further analysis by EDS detected a small amount of V and Ni inside the nano-onion, verifying the catalytic effect of these elements (Figure 13f). It should be noted that the d-orbital of the Ni element with the partially filled atomic structure can adsorb the hydrocarbon molecules, activate the dehydrogenation process, and create planar sheets of graphene [131], while VN protects the film from thermal degradations [132]. On the other hand, the detected agglomerated debris comprises oxide compounds, including $NiO_2$, $V_2O_3$, and $\alpha$-$Fe_2O_3$ nanoparticles. The defect of their surrounding graphite structure compared to the carbon onion and graphene is illustrated in Figure 14G with the $sp^2$ portion of $58 \pm 2\%$.

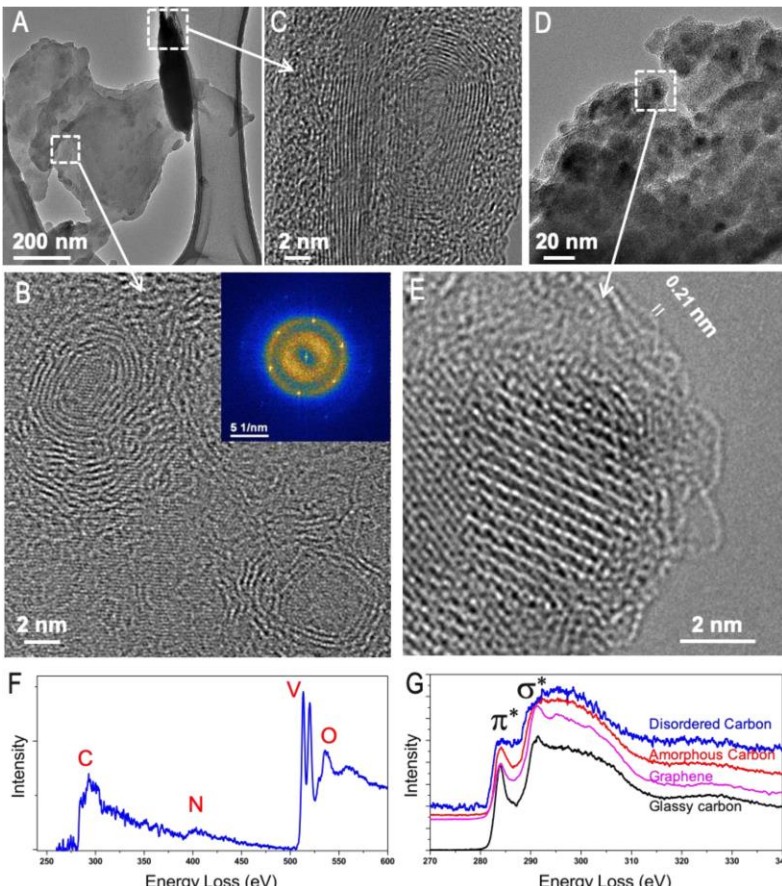

**Figure 14.** Characterized outcomes of the collected wear particles around the contact areas via TEM (**A–E**) and EELS (**F,G**) analyses [129]. Reprinted with permission and Copyright from American Chemical Society 2020.

*4.4. Mechanical Analysis*

　　To examine the mechanical properties of the tribofilms or deposited thin coatings, nanoindentation and nanoscratching can be considered as the most typical approaches. In particular, the nanoindentation experiment is conducted by applying the normal force through an indenter on the investigated specimens [133]. From that, the elastic (or Young's) modulus (E) and hardness (H) of the tribolayer and/or deposited coating can be obtained. Meanwhile, the nanoscratch one is carried out where the indented punch is laterally ploughing across the experimental surface [134]. Accordingly, the adhesion and/or interfacial characteristics of the studied sample can be evaluated. One of the DLC film's problems comes from its internal residual stresses, which result in the easy-delamination characteristic (or the low adhesion strength) [64,66,135]. Particularly, it has been detected that this coating's adhesion strength is inversely proportional to the residual stress magnitude [136]. It is clear from Donnet et al. [48] that internal compressive stress is generated during the deposition processes. Its magnitude is attributed to the kinetic energy of sputtered species interacting within the film. There are three residual stress contributions: intrinsic stress, extrinsic stress, and thermal stress. Among these, intrinsic stress is usually highlighted as the main contributor promoting the overall residual pressure. It directly relates to the films' morphology and depends on the deposition parameters, which determine the striking particles' kinetic energy within the film during the sputtering process. The physical basis that can quantify the intrinsic stress can be found in the works performed by Windischmann et al. [137] and Davis et al. [138]. Meanwhile, such external impurity atoms as oxygen and hydrogen also exert considerable influences on the microstructure of the resultant films. They promote lattice distortions that can decrease grain energy, resulting in volume expansion and allowing for the development of internal compressive stress in terms of extrinsic contributors [139]. To determine the overall residual stress magnitude of the deposited DLC, the methods based on the measurement of film-substrate structures' deformation have been indicated as the most conventional approaches. Their basis is significantly related to the fundament of beam theory [140]. In particular, the deviation of the X-ray diffraction (XRD) peak will be utilized to calculate the residual stress ($\sigma$) of the deposited film on such crystal substrate as a silicon wafer by the following equation:

$$\sigma = -\left(\frac{E_S}{1 - \upsilon_S}\right)\left(\frac{d - d_0}{d_0}\right) \tag{6}$$

where d and $d_0$ represent the crystal planes' interplanar distances with and without the presence of stress, respectively, while $\upsilon_s$ and $E_s$ are the substrate's Poisson's ratio and Young's modulus, respectively. Besides the peak shift in XRD spectra, the change in G-peak location in the Raman signal can also be used to determine the residual stress of DLC coating based on the following formula:

$$\sigma = 2G_f\left(\frac{1 + \upsilon_f}{1 - \upsilon_f}\right)\frac{\Delta\omega}{\omega_0} = \frac{E_f}{1 - \upsilon_f}\frac{\Delta\omega}{\omega_0} \tag{7}$$

where $\upsilon_f$, $G_f$, and $E_f$ are the film's Poisson's ratio, shear modulus, and Young's modulus, respectively. Meanwhile, $\Delta\omega = \omega_f - \omega_0$ represents the shift of the G peak in the resultant DLC film compared to the reference value $\omega_0 = 1580$ cm$^{-1}$.

　　For instance, the chromium (Cr) layer as a metallic intermediate, which is well-known for its ability to improve the DLC film absorption [66], was investigated by Shahsavari et al. [141]. In this work, different Cr interlayer thicknesses, including 10 nm, 20 nm, 40 nm, and 80 nm, were predeposited to the p-silicon via the direct current sputtering method (denoted as Cr1, Cr2, Cr3, and Cr4, respectively), before being coated by the DLC layers from CH$_4$ gas using the PECVD approach. The mechanical properties of the DLC films were examined by the nanoscratch and nanoindentation experiments using the Berkovich indenter in the Hysitron Nanomechanical instrument. The applied normal load in the nanoscratch tests was controlled to go up to 3000 μN within a 30 s

duration and a 4 μm scratching distance. The outcomes of the scratching experiment can be demonstrated in Figure 15. Accordingly, the plastic deformations of the DLC layers on Cr3 and Cr4 samples were detected as shown by the blister formations that indicate a low adhesion strength. Their friction coefficients were 0.15 and 0.12, respectively. In contrast, outstanding elastic behaviours can be observed on Cr1 and Cr2 intermediate, where no delamination owing to tip penetration was found. Their calculated H/E ratios (from the 500-μN max load nanoindentation test) were 0.10 and 0.11, respectively, which dedicates a slightly more elastic property of DLC film on Cr2 compared to the one on Cr1. This was further confirmed by their determined residual stress using equation (7), in which the magnitude of Cr2′s film with 213 GPa Young's modulus and 1589 cm$^{-1}$ G peak shift was just 1.37 GPa. Meanwhile, the value of Cr1′s film with 166 GPa Young's modulus and 1593 cm$^{-1}$ G peak shift was 1.59 GPa. Combined with prior surface analysis, these results successfully highlighted the influences of Cr interlayer roughness, as well as its nanoparticle distribution on the resultant mechanical behaviours of deposited DLC coating. Particularly, sputtering on the Cr2 interlayer, which has the lowest roughness and a superior homogeneity, resulted in the most durable DLC structure.

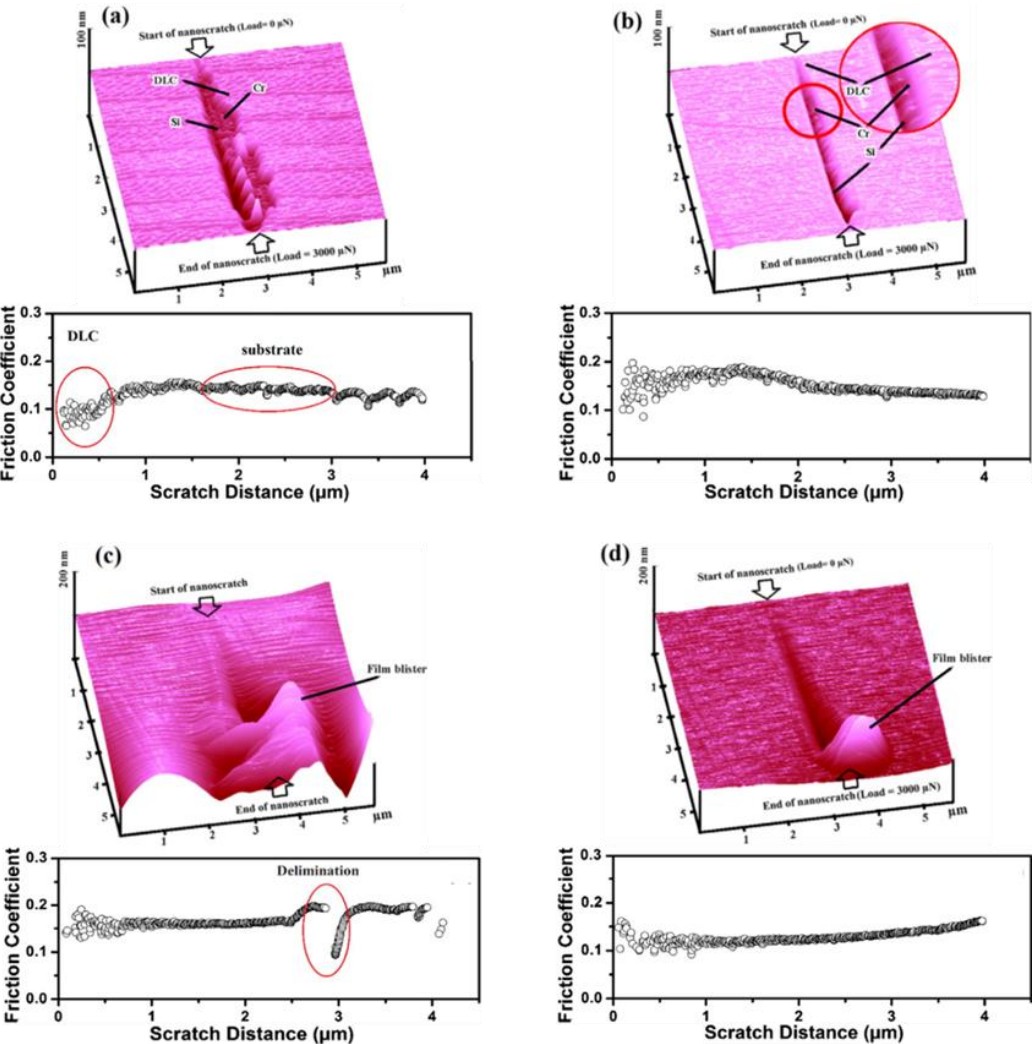

**Figure 15.** Scratching morphologies and their corresponding friction coefficient curves of the DLC film on the 10 nm (**a**), 20 nm (**b**), 40 nm (**c**), and 80 nm (**d**) thick Cr interlayer [141]. Reprinted with permission and Copyright from Elsevier 2016.

It should be noted that mechanical analysis of the in situ carbon-based tribofilms by nanoindentation is much more challenging than the deposited coatings, as the majority

of the tribofilms are nanometre-thin, which may cause an influence of the substrate on the obtained results. Thin tribofilms require careful investigation and examination of the penetration depth as well as peak load to avoid indenting more than 10% of the tribofilm thickness [142,143]. Despite this difficulty, few recent studies have implemented nanoindentation in the context of nanoscratch or nanofriction to qualitatively examine the nanoscale mechanical properties of the surface carbon-based tribofilms [35,109]. In the study by Rouhani et al. [35], a long-shaft Berkovich three-sided pyramid diamond indenter specially designed for the xSol stage was used for nanoscratch studies with a ramping load function from 0 to 500 µN at different temperatures. The scratch test at 250 °C showed the self-healing of the surface films, and the authors believe that it was due to rapid in-filling by newly formed a-C:H from palm oils. The hardness of the tribofilms was also measured with a similar tip, using a single-indentation quasi-static load function; however, the obtained hardness is similar to steel, which indicates the significant effect of the substrate on the results. In a later study by Huynh et al. [109], nanofriction was used to examine the surface tribofilms using the 150 nm diameter Berkovich diamond tip under constant load scratching mode at 250 µN and a sliding speed of 0.67 µm/s. The nanofriction curves obtained in this study show a distinct discrepancy between the areas containing carbon-based tribofilms and the bare steel areas. In addition, the higher graphitic degree of the carbon-based tribofilms in the system containing catalytic oxide tribolayers was also recognized by lower-friction behaviour at high temperatures.

## 5. Compatibility of Carbon Tribofilms with Other Lubricant Additives

Progress in the research of carbon-based tribofilm formation at the interface has raised the question about their compatibility with other additives, as formulated lubricants often comprise multiple additives for multiple purposes apart from friction and wear protection. During the development progress of DLC films for engine oil systems, there have been several studies focusing on the compatibility between these carbon-containing hard coatings and the additives within the commercial oils [144–149]. Since carbon-based tribofilms comprise predominantly carbon and hydrogen, it is expected that their behaviour will be similar to DLC films.

The interaction with the resultant ZDDP tribolayer [150–156] is usually interesting since this is a conventional antiwear additive that has been employed in the majority of practical engine lubricants [27]. In particular, the ZDDP-derived film may interact well with lower than 20% H content DLC coatings, and/or the ones containing such functional dopants as W and Si. For instance, Ren et al. [157] indicated a synergistic effect between a-C:WC coating and ZDDP tribofilm in the reciprocating tribotest at 180 °C. Meanwhile, Ruiz et al. [158] probed the performance of the resultant ZDDP tribolayer on the ta-C surface by tailoring the mechanical properties of the deposited coating. Despite the potential combination with DLC tribolayers, our current studies have reported the competitive effects between ZDDP and carbon-containing additives. In the work performed by Pham et al. [119], GNP as a modern friction modifier with the 2D layered structure [20] was blended with Amour 10W-30 commercial engine oil to investigate its interaction with existing ZDDP. The tribological results indicated that low-concentration graphene (0.05 wt% or lower) lubricants will enhance wear alleviation, but friction will be increased. In contrast, a concentration of graphene higher than 0.05 wt% will promote the tendency of stacked GNP in the tribofilm, which improves the friction reduction, but the wear will increase remarkably because of massive fine GNP particle aggregation, which destabilizes the tribofilm integrity. Additionally, the tribolayer of GNP was found to inhibit the full development of ZDDP film, evidenced by two separated regions of resultant tribofilms and the reduction in ZDDP layer thickness in the carbon-rich area (Figure 16). A similar issue was also detected when ZDDP and CPCa were blended together in the PAO4 oil lubricant [122]. Particularly, the ZDDP tribolayer produced from this combination exhibited a more discrete formation compared to the one of ZDDP alone. Along with the severe

removal of carbon-based tribofilm from CPCa, the tribotest of this combination resulted in a continuous increase in the frictional outcome and an unstable antiwear performance.

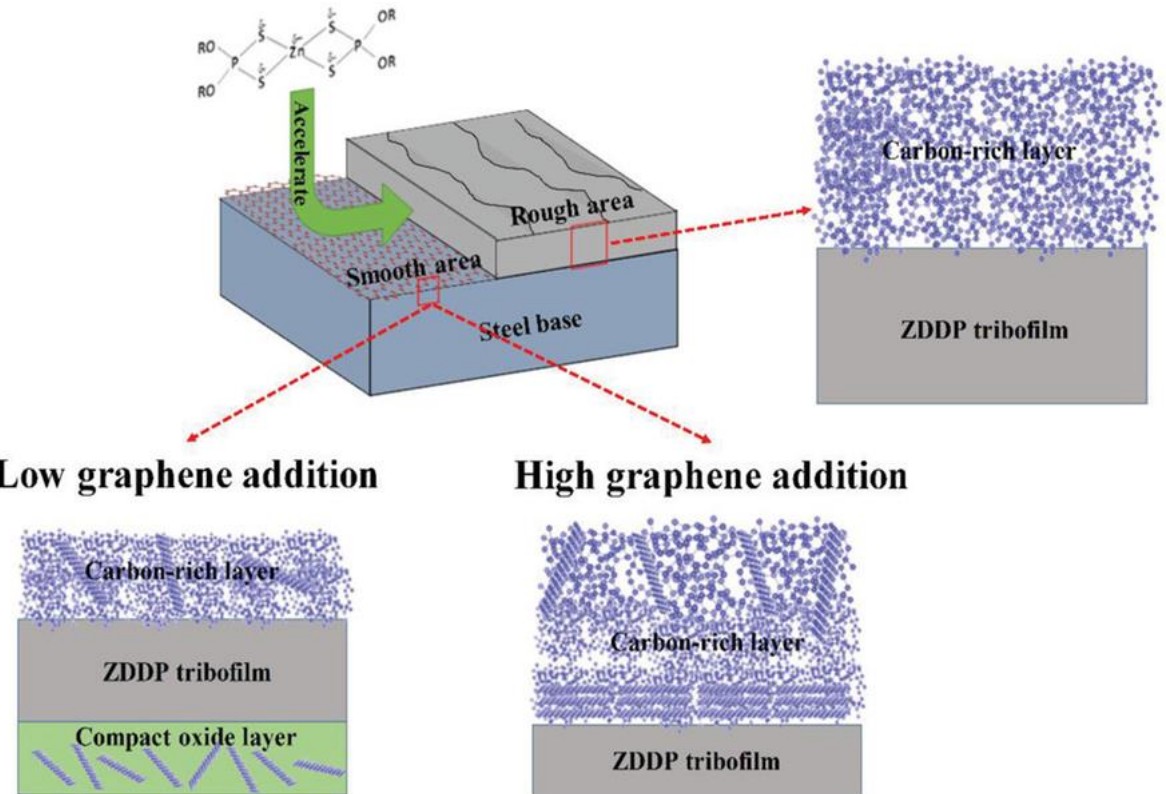

**Figure 16.** Schematic of hierarchical tribofilm structures showing the competitive effect of graphite nanoplates on the formation of the ZDDP tribofilms [119]. Reprinted with permission and Copyright from John Wiley and Sons 2019.

Adapting to the modern trends of low-viscosity lubricant development where the phosphorus and sulphur compounds are minimized [29], the DLC coatings were been studied to optimize their interactions with other "green" additives like Glycerol Mono-Oleate [159–161], fatty acid [162–165], etc. Noticeably, ultralow-friction outcomes originating from organic molecules and DLC coating interaction can be achieved by probing their hydroxyl-terminated tribolayers [166,167]. For example, Long et al. [168] reported a superlubricity friction outcome (0.004) along with a negligible wear rate by sliding an AISI52100-steel ball against the ta-C coating in the glycerol medium. In this work, the reciprocating tribotest was carried out under 386 MPa Hertz pressure at 50 °C. Based on the analysis results using XPS and MD simulation, the ultralow frictional outcome was attributed to the formation of a one-nanometre-thick film where the formation of iron oxy-hydroxide (FeOOH) termination was found on both contacting surfaces. These terminations are found to combine with water, glycerol, and/or decomposed glycerol molecules to promote a functional fluid film, guaranteeing elastohydrodynamic lubrication during the sliding process. A schematic demonstration of this fluid film can be found in Figure 17.

At the same time, inspired by the work performed by Erdemir et al. [36], the employment of lubricating nanomaterials containing active elements in promoting and managing the performances of in situ carbon-based tribolayers is showing promise. Similar to Ni, Cu nanoparticles are already famous for their outstanding lubricating properties [169]. It is worth noticing that these nanoparticles have been found to exhibit a better synergistic effect on the DLC coatings compared to those derived from ZDDP [170]. Especially, superior catalytic and tribological properties of Cu to Ni were reported when these metallic particles were blended with CPCa in the PAO4 oil lubricant [42]. Apart from our LDH work [109],

the interest in active metallic particles also spread to the research of other lubricating 2D additives. This can be evidenced by the work performed by Tang et al. [171], where the catalytic Ag was dotted on the surfaces of black phosphorus to improve the performance of this additive in PAO6 oil [172,173]. Meanwhile, besides the aforementioned combination, the carbon-based tribolayer dissociated from CPCa was also found to be compatible with the TiN coating, resulting in superlubricity outcomes [38].

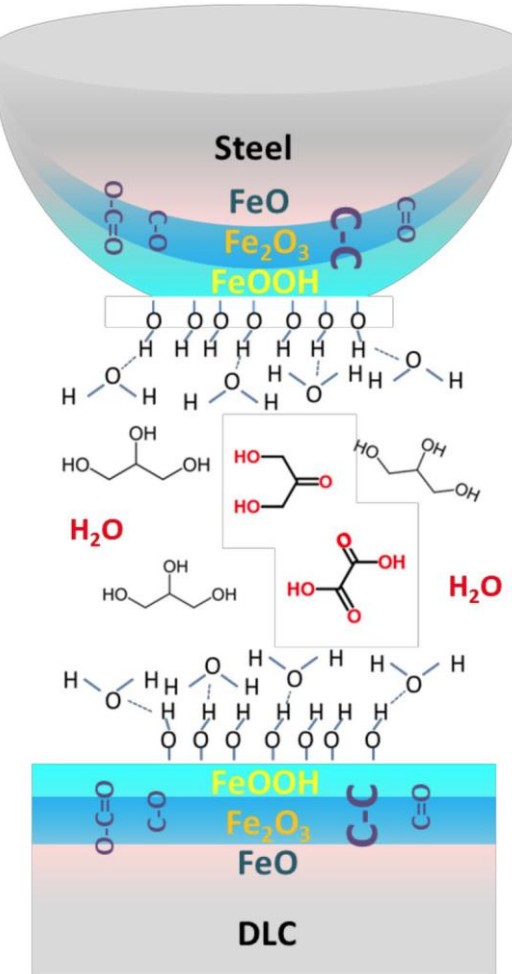

**Figure 17.** Schematic demonstration of the resultant fluid film promoting the superlubricity sliding outcomes when the AISI52100-steel ball is slid against the ta-C coating [168].

In general, it is clear that the employment of active agents has played a key role in the interaction management between carbon-based tribofilm and other additives. Each agent has exerted a unique influence on the resultant performance of the experimental carbon-based tribolayer. As a result, to employ the lubricating property of carbon-based tribofilm in practical applications, future research should focus on developing new additive packages, where such phosphorus- and/or sulphur-based additives as ZDDP are completely removed. However, with the emergence of new lubricant additives, complex interactions may occur, leading to a larger number of additives being added when formulating new additive packages. Larger additives are often required when the antagonistic effect happens between the lubricant additives to compensate for the deficiencies of new base fluid alternatives. Thus, studies on controlling interactions among additives are expected to become even more challenging. This is where the prospect of developing multifunctional additives comes as a game-changer. Advances in multifunctional lubricant additives are progressing, which are driven not only by new materials/molecules [174] and improved properties of

existing materials by functionalization [175], but also by new approaches to combine the functional materials [176].

## 6. Outlooks

Several studies in recent years have demonstrated success in controlling the formation of carbon-containing tribofilms by either chemical/physical vapour deposition or tribochemical reactions. Due to its high potential for sustainable lubrication, it is expected that studies in carbon-based tribofilms will flourish in the next few years. While optimizing and controlling the formation of carbon-containing tribofilms by different approaches will be still the main focus, the promise of a self-replenishing capability utilizing different hydrocarbon sources will open a new pathway to utilize these carbon-based tribofilms in different lubrication conditions. In a very recent review, Berman et al. [26] outlined the potential applications of carbon tribofilms formed by tribocatalysis in different fields. It should be noted that the properties of the resultant carbon-based tribofilms vary greatly depending on the lubricant formulations, hydrocarbon base, and lubricating conditions. These parameters affect the chemical structures at the nanoscale of the carbon-based tribofilms, which can have a strong positive or negative effect on the performances, as also claimed by Berman et al. [26]. Yet, the nanoscale tribochemistry of such carbon-based tribofilms in relation to their mechanical properties and macroscale performances remains largely "terra incognita".

A rich literature has been devoted to understanding the chemical structures and lubrication mechanism of DLC films, but this is still a remaining question with the in situ formed carbon-based tribofilms since these films are distinctively different in their physicochemical nature [40]. Most conclusions about the structural characteristics and lubrication mechanisms of the carbon-based tribofilms are often drawn from macroscopic observation of the wear surfaces as well as a combination of analytical techniques, including Raman, micro-Fourier transform infrared spectroscopy (micro-FTIR), nuclear magnetic resonance (NMR) and mass spectrometry (MS), diffraction analysis, and reactive MD simulations. However, they can only provide limited information on the chemical and crystalline structures. The natural questions to ask are whether these carbon-based tribofilms can be ascribed to be graphitic, diamond-like, polymeric, or something else (e.g., paracrystalline or turbostratic carbons), and how these tribofilms locally bind on different substrates. Local and interfacial structures at the nanoscale are often a minor component of the ensemble but are locally the major phase, which require spatially resolved analytical techniques to uniquely address these questions. Electron microscopy can provide this unique capability to address these questions; however, the dose–damage constraint is still a big issue in preserving the local structures of these carbon-based materials. To have far-reaching implications of carbon-based tribofilms in practice, it is essential to understand their intrinsic structure–property relationship as well as the growth rate kinetics of these films under different conditions. This information has not yet been clearly answered up to now, and it is expected that more fundamental studies will be established in the near future.

## 7. Conclusions

This study has reviewed some of the current state-of-the-art research dealing with in situ-generated carbon-based tribofilms at sliding surfaces. Particularly, we have briefly described the formation approaches as well as the performance of current carbon-based tribofilms, highlighted the key differences and/or advantages of each, and provided key insights into the current understanding of the underlying formation mechanisms as well as the most common analysis techniques alongside the potential interactions of the Carbon-based tribofilms with other additives. It is clear that in situ carbon-based tribolayers are exhibiting great potential in terms of lubrication performance and sustainability, where a lot of their benefits could be brought to develop a strategy for sustainable lubrications. However, it is necessary to consider the arising issues related to their compatibility with conventional lubricating oil packages, as well as their adaptability to different working

environments and conditions. More importantly, there are still a lot of questions and controversies about the chemical and structural properties that require advances in new analytical techniques to resolve adequately. As a result, further investigations on analysis approaches along with the formation and optimization of carbon-based tribofilms, are highly recommended to harness the full potential of such sustainable tribofilms in practical applications toward net zero emissions targets.

**Author Contributions:** Conceptualization, K.K.H. and S.T.P.; validation, S.T.P. and K.A.T.; formal analysis, K.K.H.; investigation, K.K.H. and S.T.P.; resources, K.A.T.; writing—original draft preparation, K.K.H. and S.T.P.; writing—review and editing, K.K.H., S.T.P., K.A.T. and S.W.; visualization, K.K.H. and S.T.P.; supervision, S.T.P. and K.A.T.; project administration, K.A.T.; funding acquisition, K.A.T. All authors have read and agreed to the published version of the manuscript.

**Funding:** This research was funded by Australian Research Council (ARC) grant number DP190103455 and LP160101871.

**Data Availability Statement:** Data available in a publicly accessible repositoryThe data presented in this study are openly available in published articles.

**Acknowledgments:** The study is funded by the Australian Research Council (ARC) Discovery Project DP190103455 and Linkage Project LP160101871.

**Conflicts of Interest:** The authors declare no conflict of interest.

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
