# Peer review of "Tribocatalysis Induced Carbon-Based Tribofilms—An Emerging Tribological Approach for Sustainable Lubrications"

_lubricants, doi:10.3390/lubricants11080327_

Round 1
Reviewer 1 Report
This article presents a thorough review of the most recent advancements in research on carbon-based tribofilms. The manuscript provides a comprehensive overview of the methods used for their formation, their performance characteristics, and the techniques employed to characterize them. Moreover, the article explores the interaction between tribofilms and additives. In my view, this review paper is significant and technically sound.
Some sentences are lengthy and contain multiple ideas, making it challenging to understand the intended meaning. Breaking them down into shorter, more concise sentences would improve clarity.
Author Response
The authors thank you for the comments from the reviewer. Some sentences have been shortened to improve clarity.
Reviewer 2 Report
You have done a very extensive research. Congratulations on this work. The presentation is successful. It is to be seen restrictively that you have evaluated predominantly experimental investigations. The evaluation is successful and it is helpful that you show open questions.
As a recommendation for further work, you could include more theoretical work on self-organization of tribological systems. This will show completely new aspects.
I recommend for publication
Author Response
The authors acknowledge the comments from the reviewer and we thank you for the valuable suggestions for future studies.
Reviewer 3 Report
1-Graphene or Graphite is the only consideration for this review article, or MWCNT/SMWNT are also considered?
2-I don't think CO2 emission from ICE is a point of concern here.
3-How you drive Figure 1, gather this data from which publisher?
4-For heading 2.1 and 2.2, the authors can include the physical processes details of DLC and Tribofilms Produced by Catalytic Coatings. Like what are the common techniques and methods available to prepare these coatings?
5-The coefficient of friction is reported for multiple types of coatings like for heading 2.1, and 2.2, however, the wear rate was not reported. While reported for heading 2.3, any particular reason?
6-Is this correct "Carbon-based tribofilms have been produced via the 4-ball tribo-experiment from different base oils, including pure palm oil (PO), palm oil formulation (POF), mineral oil (MO), and mineral oil formulation (MOF)", they have prepared the carbon-based tribo film via 4 ball tribo experiment or they have checked the layer behavior through 4 balls? Because in Figure 7, It is written welded, kindly check.
7- Any particular reason for including paragraphs on Ni nano-particles under heading 2.4, therefore the article is decided to carbon-based type coatings.
8- I think the heading 2.4, Mechanical Analysis is quite vague. Here you need to address the mechanical behavior of the developed carbon-based layer onto the substrate. You have started with the nanoindentation for assessing the mechanical behavior but have not built a relation from the articles in the direction of mechanical behavior.
9- One aspect is completely missing from the review, as the title suggested the sliding interface of carbon-based tribo film on the substrate; the wear behavior in terms of wear types i.e. adhesion, sliding, etc. or the stages of the wear developed against the tribometer testing on carbon-based film is completely missing. This is an important aspect as you are addressing the effectiveness of carbon-based film over the others.
minor changes required
Author Response
The authors acknowledge the comments from the reviewer. Please find our response in the attached file.

Round 2
Reviewer 3 Report
1-So in this regards, you need to add the word tribocatalysis process in the title.
2- For point 2, yes the main concern for ICE is the CO2 emission, are you going to propose a review article where carbon based tribo film were used in ICE engines as a lubricant? or any relation of carbon-based film with increase the increment or decrement of CO2 emission.
3- For point 6, so kindly correct; they "The tribological tests were carried out using 4-ball tests to evaluate the boundary lubrication performance" while previously it was written ""Carbon-based tribofilms have been produced via the 4-ball tribo-experiment from.........."
minor changes
Author Response

(The authors gave the same response as above.)
